**Smaller global and regional carbon emissions from gross land use change when considering sub-**
**grid secondary land cohorts in a global dynamic vegetation model**
Chao Yue, Philippe Ciais, Wei Li
Laboratoire des Sciences du Climat et de l'Environnement, LSCE/IPSL, CEA-CNRS-UVSQ, Université
Paris-Saclay, F-91191 Gif-sur-Yvette, France
Corresponding author: Chao Yue, chao.yue@lsce.ipsl.fr
Running title: Land use carbon emissions with sub-grid land cohorts
**Abstract**
Several modeling studies reported elevated carbon emissions from historical land use change ($E_{LUC}$) by
including bi-directional transitions on the sub-grid scale (termed gross land use change), dominated by
shifting cultivation and other land turnover processes. This has implications on the estimation of so-called
residual land $CO_2$ sink over undisturbed lands. However, most dynamic global vegetation models
(DGVM) having implemented gross land use change either do not account for sub-grid secondary lands,
or often have only a single secondary land tile over a model grid cell and thus cannot account for rotation
lengths in shifting cultivation and associated secondary forest age dynamics. Therefore it remains
uncertain how realistic the past $E_{LUC}$ estimations are and how estimated $E_{LUC}$ will differ between the two
modeling approaches with and without multiple sub-grid secondary land cohorts — in particular
secondary forest cohorts. Here we investigated the effects on historical $E_{LUC}$ over 1501–2005 by including
sub-grid forest age dynamics in a DGVM. We run two simulations, one with no secondary forests ($S_{ageless}$)
and the other with sub-grid secondary forests of 6 age classes whose demography is driven by historical
land use change ($S_{age}$). Estimated global $E_{LUC}$ for 1501–2005 are 176 Pg C in $S_{age}$ compared to 197 Pg C
in $S_{ageless}$. The lower emissions in $S_{age}$ arise mainly from shifting cultivation in the tropics under an
assumed constant rotation length of 15 years, being of 27 Pg C in $S_{age}$ in contrast to 46 Pg C in $S_{ageless}$.
Estimated cumulative $E_{LUC}$ from wood harvest in the $S_{age}$ simulation (31 Pg C) are however slightly
higher than $S_{ageless}$ (27 Pg C) when the model is forced by reconstructed harvested areas, because
secondary forests targeted in $S_{age}$ for harvest priority are insufficient to meet the prescribed harvest area,
leading to wood harvest being dominated by old primary forests. An alternative approach to quantify
wood harvest $E_{LUC}$, where it is always the close-to-mature forests that are assumed to be harvested in both
simulations, yield similar values of 33 Pg C from both simulations. The lower $E_{LUC}$ from shifting
cultivation in $S_{age}$ simulations depends on the pre-defined forest clearing priority rules in the model and
the assumed rotation length. A set of sensitivity model runs over Africa reveal that a longer rotation
length over historical period likely results in higher emissions. Our results highlight that although gross
land use change as a former missing emission component is included by a growing number of DGVMs,
its contribution to overall $E_{LUC}$ remains uncertain and tends to be overestimated when models ignore sub-
grid secondary forests.

Keywords: gross land use change, carbon emission, secondary forests, shifting cultivation, wood harvest.

**Nomenclature**
LUC : land use change
$E_{LUC}$ : carbon emissions from land use change. Positive values indicate that LUC has a net effect of
releasing carbon from vegetation to the atmosphere, while a negative value indicates the reverse, i.e.,
carbon is uptaken from the atmosphere to vegetation.
$E_{LUC\ process[,\ configuration]}$ : carbon emissions from a certain LUC *process* (***net** transitions only, land **turnover**,*
*wood **harvest** or **all** three processes combined*) quantified by a specific model *configuration* (***age** or*
***ageless***, in which differently aged sub-grid land cohorts are, or are not explicitly represented,
respectively). For instance, $E_{LUC\ net,\ ageless}$ indicates $E_{LUC}$ from net transitions only as simulated by model
runs that do not explicitly represent sub-grid age dynamics, i.e., a single ageless mature patch is used to
represent a land cover type; $E_{LUC\ net,\ age}$ indicates $E_{LUC}$ from the same process using a model configuration
that explicitly represents differently aged land cohorts since their establishment.
$S_{age}$: Model simulations that represents sub-grid secondary land cohorts since their establishment.
$S_{ageless}$: Model simulations that do not include sub-grid age dynamics, i.e., a single ageless mature patch is
used to represent a land cover type.

**1 Introduction**
Historical land use change (LUC), such as the permanent establishment of agricultural land on forests
(deforestation), shifting cultivation and wood harvest, has contributed significantly to the atmospheric
$CO_2$ increase, in particular since industrialization (Houghton, 2003; Le Quéré et al., 2016; Pongratz et al.,
2009). Carbon emissions from land use change ($E_{LUC}$) are often defined as a net effect between carbon
release on newly disturbed lands, given that in most cases newly created lands have a lower carbon
density than natural ecosystems (e.g., deforestation or forest degradation), and carbon uptake by
recovering ecosystems (e.g., cropland abandonment or afforestation/reforestation). As the high spatial
heterogeneity of land conversions precludes any direct measurement of global or regional $E_{LUC}$, modeling
turned out to be the only approach to its quantification (Gasser and Ciais, 2013; Hansis et al., 2015;
Houghton, 1999, 2003; Piao et al., 2009b). Methods to quantify $E_{LUC}$ could fall broadly into three
categories, namely bookkeeping models (Gasser and Ciais, 2013; Hansis et al., 2015; Houghton, 2003),
dynamic global vegetation models (Shevliakova et al., 2009; Stocker et al., 2014; Wilkenskjeld et al.,
2014; Yang et al., 2010), and fire-based estimates of deforestation fluxes (van der Werf et al., 2010).

When including sub-grid scale bi-directional gross land use changes such as shifting cultivation or other
forms of land turnover processes, models are found to yield higher estimates of $E_{LUC}$ for 1850-2005
ranging 2-38% depending on different models and assumptions than accounting for net transitions only
(Hansis et al., 2015). Wood harvest, although it does not change the underlying land cover type, can also
lead to additional carbon emissions due to the fast carbon release from recently harvested forests and slow
uptake from re-growing ones (Shevliakova et al., 2009; Stocker et al., 2014). Because of the importance
of these processes in understanding historical LUC emissions, gross land use change and wood harvest
have been implemented in several dynamic global vegetation models (DGVMs), as synthesized in the
Table 1 of Yue et al. (2017). A recent synthesis study by Arneth et al. (2017) reported consistent increase
in LUC emissions by several models when including shifting cultivation and wood harvest, as well as
other agricultural management processes such as pasture harvest and cropland management. These
processes altogether yield an upward shift in estimated historical LUC emissions, implying a larger
potential in the land-based mitigation in the future if deforestation or forest degradation can be stopped.

While replacing forest with cropland or pasture typically leads to carbon release, afforestation and forest
regrowth following harvest and agricultural abandonment sequester carbon in growing biomass stocks.
Some recent studies, both on site (Poorter et al., 2016) and regional scales (Chazdon et al., 2016), show
that secondary forests recovering from historical land use change are contributing to terrestrial carbon
uptake, and that the carbon stored per unit land sometimes exceeds that of the original primary forest
(Poorter et al., 2016). While explicit representing sub-grid secondary forests and other lands with
different time lengths since the last disturbance (defined as cohorts or age classes) is relatively
straightforward in bookkeeping models (Hansis et al., 2015), and is fairly easy in some DGVMs
combined with a forest gap model (e.g., LPJ-GUESS, Bayer et al., 2017), only a few DGVMs following a
"area-based" approach (Smith et al., 2001) have included sub-grid secondary lands usually with only a
single cohort for a given vegetation type. Shevliakova et al. (2009) pioneered the inclusion of both gross
land use change and secondary lands in a DGVM. Their model can contain up to a total number of 12
secondary land cohorts over a model grid cell, but the spatial separation of different natural plant
functional types (PFTs) was limited. In some other DGVMs (Kato et al., 2013; Stocker et al., 2014; Yang
et al., 2010), secondary land was limited to one cohort per PFT over a model grid cell. This has limited
the accurate representation of the carbon balance of differently aged secondary forests.

In reality, shifting cultivation and wood harvest (forestry) tend to have certain rotation lengths (McGrath
et al., 2015; van Vliet et al., 2012), which vary among different regions and forest management systems.
Simulating these LUC activities by targeting forests with an appropriate age in the model can have
important consequences in derived LUC emissions, since young versus old forests have strong difference
in aboveground biomass stocks. Using a book-keeping model, Hansis et al. (2015) showed that assuming
only secondary land clearing in gross change yields only a 2% increase in $E_{LUC}$ compared with accounting
for net transitions only, much smaller than the 24% increase when assuming primary land clearing as a
priority in gross change. In view of the fact that worldwide, systematic information on historical and
present rotation lengths of shifting cultivation and wood harvest is missing, some reconstructions of land
use change, such as the land-use harmonization version 1 (LUH1) data assumed a fixed rotation length of
15 years for shifting agriculture in the tropics, and this assumption has been used in some modeling
studies (Bayer et al., 2017).

Past studies on $E_{LUC}$ using DGVMs mainly focused on the issue of difference in LUC emissions between
accounting for gross land use change and net transitions only. Very few studies have addressed the issue
of how much $E_{LUC}$ from gross transitions differ by assuming clearing of primary forests versus secondary
forests. The former problem can be tackled by DGVMs without sub-grid secondary lands, while the latter
one can only be addressed by DGVMs with an explicit sub-grid secondary land age structure, if rotation
lengths in different regions are to be accounted for. Furthermore, it is unclear either how large is the
impact of variable shifting cultivation rotation lengths on estimated $E_{LUC}$.

In this study, we quantify global and regional carbon emissions from historical gross land use change
since 1501 using a global vegetation model ORCHIDEE (ORganizing Carbon and Hydrology In Dynamic
EcosystEms) that has recently incorporated gross land use change and wood harvest, along with the
representation of sub-grid secondary land cohorts of different ages. The model development and
examination of model behaviour on site and regional levels are documented in a companion paper (Yue et
al., 2017). The current paper focuses on its global application and quantified emissions. Our objectives
are: 1) to quantify global and regional carbon emissions from historical gross land use change since 1501
and compare them with previous studies, and to examine the impacts on $E_{LUC}$ when considering sub-grid
secondary land cohorts by using parallel model simulations with and without sub-grid secondary land
cohorts. 2) Examine contributions to $E_{LUC}$ from different LUC processes (i.e., net transitions only, shifting
cultivation or land turnover, and wood harvest) and how they differ between the two model configurations
with and without secondary land cohorts. 3) To examine the impacts on $E_{LUC}$ of different assumptions on
rotation lengths in shifting cultivation using Africa as a case study. Hereafter, we will use the terms
'shifting cultivation' or 'land turnover' interchangeably as they refer to the same process in the model —
bi-directional equal-area land transitions between two land use types.

**2 Methods**
**2.1 ORCHIDEE-MICT v8.4.2 model and the implemented gross land use change processes**
ORCHIDEE (Krinner et al., 2005) is a dynamic global vegetation model (DGVM) and the land surface
component of the IPSL Earth System Model (ESM). It comprises three sub-models operating on different
time steps. SECHIBA operates on half-hourly time step and simulates fast exchanges of energy, water
and momentum between vegetation and the atmosphere. STOMATE operates on daily time step and
simulates vegetation carbon cycle processes including photosynthate allocation, plant phenology,
vegetation mortality and recruitment. The third sub-model contains various modules about different
processes on varying time steps, such as vegetation dynamics (daily), fire disturbance (daily), and land
use change (annual time step).

The land use change module originally contained in ORCHIDEE was developed in (Piao et al., 2009a)
where only net transitions are taken into account. Recently, gross land use change and explicit
representation of differently aged sub-grid land cohorts have been developed in a branch of ORCHIDEE
model known as ORCHIDEE-MICT (Guimberteau et al., 2017). This model will be henceforth referred to
as ORCHIDEE-MICT v8.4.2 (Yue et al., 2017). Idealized site-scale simulations with this model have
shown that estimated carbon emissions from shifting cultivation and wood harvest are reduced by
explicitly including sub-grid age dynamics, in comparison with an alternative approach to representing
land cover types with a single ageless patch. This is because the secondary forests that are cleared in
shifting cultivation or wood harvest with a rotation length of 15 years have lower biomass than the forests
in the ageless parameterization, which have carbon stocks close to mature forests. Yue et al. (2017)
provides details on the processes involved in explaining differences in $E_{LUC}$ regarding whether sub-grid
forest age structure is considered or not.

The gross land use change module in ORCHIDEE-MICT v8.4.2 operates on an annual time step. For the
very first year of the simulation, an initial land cover map (represented as a map of plant function types or
PFTs) is prescribed. Land cover maps of following years are updated annually using land use transition
matrices corresponding to LUC processes. Land use transitions among four vegetated land cover types are
included: forest, natural grassland, pasture and cropland. The model separates overall LUC into three
additive sub-processes in order to diagnose their individual contributions to $E_{LUC}$, namely net land use
change equivalent to the original approach that considers net transitions only, land turnover equivalent to
shifting cultivation, and wood harvest. Matrices for net land use change and land turnover ($[X_{i,j}]$) take the
form of 4 rows by 4 columns, with $X_{i,j}$ indicating the land transition from vegetation type $i$ to $j$. The
matrix for wood harvest has only two elements, indicating ground fractions of forest subject to harvest
from primary and secondary forests, respectively. The current model version assumes that bare land
fraction remains constant throughout the entire simulation.

As is mentioned above, ORCHIDEE-MICT v8.4.2 is capable of representing sub-grid secondary even-
aged land cohorts or age classes, expressed to have different time lengths since their establishment.
Differentiation of age classes applies on all vegetation types in the model. The number of age classes for
each PFT can be customized via a configuration file. Age classes for forest PFTs are distinguished in
terms of woody biomass, while those for herbaceous PFTs are defined using soil carbon stock. Newly
transitioned land is assigned to the youngest age class. Forest cohorts will move to the next age class
when their woody biomass exceeds the threshold during forest growth. For herbaceous PFTs, younger age
classes are parameterized to have a smaller soil carbon stock. This serves mainly as a preliminary attempt
to have cohorts of secondary lands for herbaceous vegetation. Because the directional change of soil
carbon largely depends on the vegetation types before and after LUC and on climate conditions (Don et
al., 2011; Poeplau et al., 2011), ideally agricultural cohorts from different origins (and age since
conversion) should be differentiated, with a origin-specific soil carbon boundary parameterization.
However, to avoid inflating the total number of cohorts and the associated computation demand, as a first
attempt here, we simply divided each herbaceous PFT into two broad sub-grid cohorts according to their
soil carbon stocks and without considering their individual origins. We expect that such a
parameterization can accommodate some typical LUC processes, such as the conversion of forest to
cropland where soil carbon usually decreases with time, but not all LUC types (for instance, soil carbon
stock increases when a forest is converted to a pasture).

To simulate land use change when taking into account sub-grid land cohorts, a set of priority rules
become necessary regarding which land cohorts to target given a specific LUC type (Table 1 in Yue et al.,
2017), and regarding how to allocate LUC area into different PFTs of the same age class. For net land use
change, clearing of forests exclusively starts from the oldest cohorts and then moves onto younger ones
until the youngest ones. For shifting cultivation or land turnover, forest clearing starts from a pre-defined
middle-aged class, and then moves onto older ones if this starting age class is used up, until the oldest

205 ones. The primary target forest cohort in shifting cultivation and wood harvest can be parameterized in

206 the model. For the current study, shifting cultivation primarily targets the $3^{rd}$ youngest cohort (Cohort$_3$)

207 and wood harvest primarily targets the $2^{nd}$ youngest cohort (Cohort$_2$), with a total number of 6 forest

208 cohorts (Cohort$_1$ to Cohort$_6$, with Cohort$_1$ being the youngest) being simulated. This is to accommodate

209 the assumption used in the LUC forcing data that shifting cultivation has a certain rotation length (see the

210 Sect. 2.2), so that secondary forests are given a high priority to be cleared for agricultural land, and older

211 forests will be cleared when even more agricultural lands are needed. Secondary forest wood harvest

212 follows the same rule as shifting cultivation regarding on which forest cohorts to clear. Finally, for all

213 other land cover types that are used as a source for conversion and primary forest harvest, we start from

214 the oldest age class and move sequentially to younger ones, in order to meet the prescribed LUC area in

215 the forcing data. After the LUC area is allocated on the cohort level, it is then distributed among different

216 PFTs in proportion to their existing areas in this cohort.

217

218 In order to compare the simulated $E_{LUC}$ with and without sub-grid secondary land cohorts, ORCHIDEE-

219 MICT v8.4.2 can be run in a way that each PFT has one single age class. This is equivalent to the

220 alternative approach by which no sub-grid land cohorts are simulated. For more information on the

221 rationale and details of LUC implementation in ORCHIDEE-MICT v8.4.2, readers are referred to Yue et

222 al. (2017).

223

224 **2.2 Preparation of forcing land use change matrices**

225 For historical land use transitions, the land use harmonized data set version 1 (LUH1) for the CMIP5

226 project was used (Hurtt et al., 2011, http://luh.umd.edu/data.shtml#LUH1_Data). We used the version of

227 LUH1 data without urban lands as ORCHIDEE-MICT v8.4.2 does not simulate the effects of urban lands.

228 The original data set is at a 0.5° spatial resolution with an annual time step covering 1500-2005. Four land

229 use types are included: primary natural land, secondary natural land, pasture and cropland. The type of

230 "natural land" consists of grassland and forest (which are separated in ORCHIDEE-MICT) but their

231 relative fractions are not separated.  In LUH1, land use transitions from either primary or secondary

232 natural land to pasture or cropland are provided, and vice versa. Secondary natural lands originated from

233 pasture or cropland abandonment. Besides, land use transitions between pasture and cropland are

234 provided as well. Harvested wood comes either from primary or secondary forest or non-forest lands,

235 with ground area fractions that are harvested being available. Note that this does not contradict with the

236 fact that forest and grassland fractions are not separated within the land use type of "natural land" in

237 LUH1, because forests are defined as natural lands with a certain biomass carbon, which is further

238 simulated by a terrestrial model (Hurtt et al., 2011).


Rather than the simple terrestrial model (Miami-LU) used in Hurtt et al. (2011) to separate natural
vegetation into forested and non-forest land, ORCHIDEE-MICT distinguishes 8 forest PFTs, 2 natural
grassland PFTs, 2 cropland PFTs (Krinner et al., 2005) and 2 pasture PFTs. Thus, to use LUH1 LUC
transition reconstructions as a forcing input, assumptions have to be made to disaggregate LUH1 land use
types into corresponding ORCHIDEE PFTs. For this purpose, we used an ORCHIDEE-compatible PFT
map generated from the European Space Agency (ESA) Climate Change Initiative (CCI) land cover map
(shortened as the ESA-CCI-LC map) covering a 5-year period of 2003-2007 (European Space Agency,
2014), assuming that it corresponds to the land use distribution for 2005 by the LUH1 data. Subsequently,
we backcast historical PFT map time series for 1500-2004 based on this 2005 PFT map using LUH1
historical net land use transitions as a constraint. Because land turnover involves an equal, bi-directional
land transition between two land cover types, it does not lead to any net annual changes in the PFT map.
Therefore, only net transition information is needed when backcasting historical PFT maps.

The guiding principle of backcasting is that when ORCHIDEE is forced by historical net land cover
matrices (as constrained by the LUH1 data) starting from the year 1500, it should reach exactly the PFT
map in 2005 based on ESA CCI land cover map. To separate land use transitions in LUH1 into processes
of net land use change and land turnover, we simply treat net land use change as the land transitions
excluding the minimum reverse fluxes between two land use types. During the backcasting process,
reconciliations have to be made where LUH1 data disagrees with the ESA map on the grid cell scale.
When backcasting historical PFT map time series using net land use change matrices, we assume that
when pasture or cropland is created, they come from an equal share of forest and grassland; when their
fractions decrease, cropland abandonment leads first to forest recovery and then followed by natural
grassland expansion, while pasture abandonment leads to an equal share of forest and natural grassland
expansion. We then treat the minimum of two reverse land fluxes between secondary natural land and
cropland or pasture as land turnover transitions. For each year, the land turnover transition between two
land use types is not allowed to exceed the minimum of their existing areas. Spatially resolved forest
harvest time series are provided in LUH1. We built the wood harvest matrices by limiting wood harvest
area within the total area of forest PFTs over each grid cell for each year. Primary and secondary forest
wood harvests from LUH1 were included and treated as primary and secondary forest harvest in the
model, respectively, with non-forest wood harvest being discarded. For more details on PFT map
backcasting and the construction of land use transition matrices, readers are referred to the Supplement
Material.

The construction of historical PFT maps and land transition matrices was done at 2° resolution for the
whole globe, after re-sampling all input data from their original resolution to 2°. The reconstructed global
forest area agrees with that by Peng et al. (2017), who has backcast historical ORCHIDEE PFT map
series using the same ESA-CCI-LC 2005 PFT map and historical pasture and crop distributions from
LUH1 but not the LUH1 land use transitions, with historical forest areas in the nine regions of the globe
being constrained by data in Houghton (2003) based on national forest area statistics. The land turnover
transitions between secondary land (forest and grassland) and cropland (or pasture) from the matrices
defined above are smaller than originally prescribed in LUH1, because some of the prescribed transitions
are ignored due to the inconsistency between LUH1 map in 2005 and the 2005 ORCHIDEE PFT map
(See Supplement Material for detailed comparison). Because of this inconsistency, around 35% of net
transitions from natural land to pasture, and 14% of net transitions from natural land to cropland were
omitted when adapting the LUH1 data set to our model. About 20% of the turnover transitions between
secondary land and pasture were omitted, and 11% of turnover transitions between secondary land and
cropland were omitted. Such inconsistencies among different data sets are a rather common challenge for
their application in DGVMs, which have been reported by, for example, in Li et al. (2017a), Meiyappan
and Jain (2012) and  Peng et al. (2017). Note that shifting cultivation (land turnover) is limited to the
tropical band as in LUH1, and the land turnover change resulting from the gridded LUH1 data upscaling
from 0.5° to 2° is not included. The missing land turnover areas represent 17% of the turnover between
natural lands and cropland that are included in our study, and 14% of the turnovers between natural lands
and pasture. The influence of this spatial aggregation error on derived emissions will be discussed in the
discussion section.

**2.3 Simulation protocol**
**2.3.1 Separate contributions of different land use change processes**
The PFT map of year 1500 as generated from the backcasting procedure (see the previous section) was
used during the model spin-up. Climate data used were CRUNCEP v5.3.2 climate forcing at 2° resolution
covering 1901-2013 (https://vesg.ipsl.upmc.fr/thredds/fileServer/store/p529viov/cruncep/readme.html).
For the spin-up, climate data were cycled from 1901 to 1910, with atmospheric $CO_2$ concentration being
fixed at the 1750 level (277 ppm). Following LUH1 (Hurtt et al., 2011), we assume that no land use
change occurs during the model spin-up. This might lead to overestimation of $E_{LUC}$ for the beginning
years of the transient simulation due to high carbon stocks that are free from LUC activities before 1501.
But on the other hand, legacy emissions from LUC activities before 1501 are also omitted. In general,
because the magnitude of annual LUC activities for 1501–1520 is very small (data shown in Fig. 2), we
assume the bias of LUC emissions induced by not including LUC in the spin-up is small. Besides,
simulated $E_{LUC}$ is less influenced by this factor after ca. 1700, which dominates the total LUC emissions
since 1501. The spin-up lasts for 450 years and includes a specific accelerated soil carbon module to
speed up the equilibrium of soil carbon stock. Fires and fire carbon emissions are simulated with a
prognostic fire module (Yue et al., 2014), with fire occurring only on forests and natural grasslands.
Simulated net land-atmosphere carbon flux is calculated as net biome production (NBP):

$$NBP = NPP - F_{Inst} - F_{Wood} - F_{HR} - F_{Fire} - F_{AH} - F_{pasture} \qquad \text{Eq (1)}$$

Where NPP is the net primary production. All fluxes starting with "F" are outward fluxes (i.e., carbon
source from the ecosystem perspective), with $F_{Inst}$ being instantaneous carbon fluxes lost during LUC
(e.g., site preparation, deforestation fires etc.), $F_{Wood}$ for delayed carbon emissions from the degradation of
harvested wood product pools, $F_{HR}$ for soil respiration, $F_{AH}$ for carbon emissions from agricultural
harvest, including harvest from croplands and pastures (treated as a carbon source for the year of harvest
equaling the harvested biomass; this source is assumed to occur on the grid cell harvested, ignoring the
transport, processing and final consumption of agricultural yield), and $F_{pasture}$ for additional non-harvest
carbon sources from pastures including export of animal milk and methane emissions. Carbon emissions
from land use change ($E_{LUC}$) are quantified as the differences in NBP between simulations without and
with LUC, with positive values representing carbon sources (i.e., LUC emissions). We conducted a set of
additive factorial simulations (S0 to S3) by including matrices of different LUC processes in each
simulation (Table 1), which allow quantifying $E_{LUC}$ from different LUC processes. Note that this
separation is done from a theoretical point of view with the objective to investigate the impacts on
quantified emissions from gross land use change when including sub-grid multiple land cohorts. The
simulations of S0 to S3 allow separating the contribution to $E_{LUC}$ by different LUC processes in a fully
additive manner and this works accurately for a linear system. To test the uncertainties in $E_{LUC\ turnover}$ and
$E_{LUC\ harvest}$ introduced by this assumption, we performed an alternative S2b simulation, which includes
both net land use change and wood harvest. $E_{LUC\ turnover}$ and $E_{LUC\ harvest}$ are then calculated using both S2
and S2b simulations and emissions from these two factorial runs are compared with each other.
Henceforth for briefness, we denote the simulation without sub-grid age class dynamics as $S_{ageless}$,
simulation with sub-grid age dynamics as $S_{age}$. At last, to investigate the sensitivity of $E_{LUC\ turnover}$ to
shifting cultivation rotation length, we performed further simulations for Africa as a case study. Another
five simulations were branched from the S2 simulation starting from the year 1860, in which the primary
target cohort for land turnover was varied as each of the five cohorts other than Cohort$_3$, the default
primary target cohort for land turnover.

**2.3.2 Define thresholds for age classes**

For the simulation with age dynamics ($S_{age}$), six age classes are used for forest PFTs and two age classes for other PFTs. As explained, age classes of forest PFTs are separated in terms of woody biomass. The LUH1 data assumes a 15-year residence time for agricultural land in shifting cultivation in tropical regions. Ideally, model parameterization of woody biomass thresholds should allow corresponding forest age being inferred, so that clearing of forest age class in the model could match that in the LUH1 data set. For this purpose, we fit a woody biomass-age curve for each forest PFT using the model data from the spin-up:

$$B = B_{max} \times [1 - \exp(-k \times age)] \qquad\qquad Eq (2)$$

where $B_{max}$ is the asymptotic maximum woody biomass; k is the biomass turnover rate (in unit of $yr^{-1}$). The curve-fitting used PFT-specific woody biomass time series during spin-up by averaging all grid cells across the globe. The ratios of thresholds of each age class to the maximum woody biomass ($B_{max}$) are looked up from this curve, based on their corresponding forest ages (Table 2). Next, these ratios are multiplied with the equilibrium woody biomass at each grid cell, to derive a spatial map of thresholds in woody biomass. We set the corresponding age for the Cohort$_3$ for tropical forests as 15 years, in line with the residence time of shifting cultivation assumed in LUH1. Considering that temperate and boreal forests grow slower than tropical ones, forest ages corresponding to the Cohort$_3$ are set as 20 and 30 years for temperate and boreal forests, respectively.

We acknowledge that using such static woody biomass boundaries cannot ensure the exactly a forest of a given age to be cleared in the transient simulation, because changes in environmental conditions (e.g., atmospheric $CO_2$ concentrations, climate) may alter the woody biomass-age curves established from the spin-up results, i.e. the boundary biomass limit is reached at a younger age in case productivity increases from environmental condition changes. If we assume that land managers always clear forest according to their ages, then our simulated land use emissions might be underestimated, provided a higher biomass for a given age in transient simulations than for the spin-up state. But in general the uncertainties of using static biomass boundaries for forest cohorts should be less influential than the uncertainty brought about by the fact that — globally, rotational lengths of land turnover are poorly known and we have assumed a constant 15-year rotation length for shifting agriculture in tropical regions. For wood harvest, we also assumed three different simple fixed rotation lengths for boreal, temperate and tropical regions, respectively (Table 2).

We used two age classes for each herbaceous PFT including natural grassland, cropland and pasture,
representing high versus low soil carbon densities, respectively. The energy balance in ORCHIDEE-
MICT v8.4.2 is resolved over the whole grid cell, and the hydrological balance is calculated over sub-grid
soil tiles (bare soil, forest and herbs) rather than over each PFT. We thus expect the factors influencing
soil carbon decomposition (i.e., soil temperature, soil moisture) to have little difference between different
age classes of the same PFTs. This justifies the small number of age classes for herbaceous PFTs selected
here as it can maximize computing efficiency. Overall, this feature of separating herbaceous MTCs into
multiple cohorts is coded more as a "place holder" for the current stage of model development rather than
having solid scientific significance. Fully tracking soil carbon stocks of different vegetation types and
their transient changes following land use change would require a much larger number of cohorts than
that used in this study.

In $S_{age}$ simulations, clearing of forest in the process of land turnover starts from Cohort$_3$ in tropics, ,
corresponding to 15 year-old forest, and forest clearing for wood harvest starts from Cohort$_2$. Wood
product pools resulting from net land use change and land turnover, and those from wood harvest are
tracked separately in the model. However, land patches created from different LUC activities are not
tracked individually, e.g., young forests, either re-established from land turnover or wood harvest, are
merged together. In this approach, it is not possible to attribute the carbon fluxes into exact LUC
processes, which explains why factorial simulations are needed to attribute contributions from different
LUC processes. Within the model, wood harvest module is executed before the modules of net land use
change and land turnover. This is reasonable as a forest might be harvested prior to being converted to
agricultural land. Last, we turned off the dynamic vegetation module as allowing dynamic vegetation and
backcasting historical land cover maps using prescribed land transitions are internally inconsistent.

**3 Results**
**3.1 Global carbon emissions with and without sub-grid age dynamics**
Simulated $E_{LUC}$ for 1501-2005 for different LUC processes and model configurations are shown in Table
3. The model simulates a cumulative $E_{LUC\,net}$ of 123.7 and 118.0 Pg C for 1501-2005, for cases of without
and with sub-grid age dynamics, respectively. Including land turnover and wood harvest yields additional
carbon emissions in both cases, with $E_{LUC\,turnover}$ as 45.4 Pg C and $E_{LUC\,harvest}$ as 27.4 Pg C in $S_{ageless}$.
Accounting for age dynamics, in contrast, generates a lower $E_{LUC\,turnover}$ of 27.3 Pg C, or 40% lower than
that obtained by the $S_{ageless}$ simulation. $E_{LUC\,harvest}$ for $S_{age}$ equals to 30.8 Pg C and is slightly higher than in
$S_{ageless}$. When wood harvest is included on top of only the net land use change (the S2b simulation), the
$E_{LUC\,harvest\,S2b}$ obtained by differing S1 and S2b simulations is slightly higher than that when wood harvest
is included as the last term (i.e., quantified by differing S2 and S3 simulations). This is reasonable
because in the latter case, forests subject to wood harvest were already under disturbances of both land
turnover and net land use change, which reduces their carbon stocks before harvest is applied on pre-
defined areas. The $E_{LUC\ turnover}$ derived from S2b simulations, in contrast, is lower than that derived from
S2 simulations (Table 3).  Nonetheless, a consistent lower $E_{LUC\ turnover}$ is obtained by accounting for sub-
grid age dynamics than not, by 40% or 37% depending on the S2 or S2b simulation being used.
Furthermore, different estimations of land turnover emissions derived by S2 and S2b simulations are
close to each other, with a difference of ~10% of their mean value, indicating that LUC emissions are
quasi-linear system with respect to the different LUC processes. Based on this and for simplicity, in the
following we will mainly focus on the results using S2 simulations.

Figure 1 shows the time series of simulated $E_{LUC,\ all}$ from all LUC processes (net land use change + land
turnover + wood harvest) in comparison with previous studies. Simulated $E_{LUC}$ from each individual LUC
process and corresponding time series of LUC areas are shown in Fig. 2, with the temporal changes in
emissions of land turnover and wood harvest by S2b simulations being shown in Fig. S7. All estimations
show a gradual increase of $E_{LUC}$ starting from the early 18[th] century with a peak of 1.5–3.5 Pg C yr$^{-1}$
around the 1950s, followed by a slight decrease during 1970s and 1980s and then another peak appeared
for 1990s. $E_{LUC}$ simulated by ORCHIDEE-MICT v8.4.2 is at the lower bound of all estimations until
1950s, but its second peak of emissions around 1990s (1.7–1.8 PgC yr$^{-1}$) is a little higher than the first one
(1.5 Pg C yr$^{-1}$). $E_{LUC\ all,\ ageless}$ remains slightly higher than $E_{LUC\ all,\ age}$ until ca. 1960, and after that the
difference increases to 0.25 Pg C yr$^{-1}$. This two-peak pattern over time in $E_{LUC\ all}$ by ORCHIDEE-MICT
v8.4.2 is mainly driven by $E_{LUC\ net}$ (Fig. 2a) which also shows two peaks around 1950s and 1990s,
consistent with the peaks of land use change areas in the LUH1 forcing data (Fig. 2d). It should also be
noted that as $E_{LUC}$ is quantified as the difference in NBP between two model simulations, its magnitude
thus depend both on the magnitude of areas subject to LUC and the magnitude of carbon fluxes in the
reference S0 simulations, as driven by climate variability, atmospheric $CO_2$, etc.

Consistent with the idealized site-scale simulation in Yue et al. (2017), $E_{LUC\ turnover,\ ageless}$ is higher than
$E_{LUC\ turnvoer,\ age}$ (Fig. 2b). Emissions from instantaneous fluxes and harvested wood product pool are lower
in the $S_{age}$ simulation than in $S_{age}$ because in the former case low-biomass secondary forests are converted
to agricultural land, as opposed to high-biomass mature forests in the latter one. Similarly, the lower land
turnover emissions in the $S_{age}$ simulation than $S_{ageless}$ are also found in the results of the S2b simulation
(Fig. S7). The difference in $E_{LUC\ turnover}$ between the two simulations explains the higher $E_{LUC\ all}$ obtained
by the $S_{ageless}$ simulation. On the other hand, $E_{LUC\ net}$ does not differ much between the two simulations
(Fig. 2a), since in both cases it is mature forests that are converted, which have little difference in their
biomass densities between $S_{ageless}$ and $S_{age}$. Both $E_{LUC\ turnover,\ ageless}$ and $E_{LUC\ turnover,\ age}$ roughly follow the
temporal pattern of areas impacted by land turnover from LUH1 (Fig. 2e), with a steep increase starting
from ca. 1900 until 1980, corresponding to a strong increase in the areas undergoing forest-pasture gross
transitions, dominated by tropical regions. After 1980 the turnover-impacted area somewhat stabilizes and
then shows a slight decrease. Accordingly, $E_{LUC\ turnover,\ ageless}$ shows only a corresponding slight decrease of
emissions in Fig. 2b, while $E_{LUC\ turnover,\ age}$ has a much bigger decrease, driven by the fact that recovering
secondary forests gain carbon quickly after being taken out of shifting agriculture systems.

Finally, $E_{LUC\ harvest}$ between $S_{age}$ and $S_{ageless}$ simulations are almost identical until 1800 (Fig. 2), during
which the wood harvest area remains stable (Fig. 2f). After this, $E_{LUC\ harvest,\ ageless}$ is lower than $E_{LUC\ harvest,}$
$_{age}$ for the 19$^{th}$ and most of the 20$^{th}$ century when $E_{LUC\ harvest}$ continued to rise, mainly driven by a rise in
secondary forest harvest area (Fig. 2f). According to the priority rules of secondary forest harvest in $S_{age}$,
older forests, until the oldest ones, will be harvested if existing young forest age classes are not sufficient
to meet the prescribed harvest target. This most likely happens when harvested area continues to rise,
simply because existing secondary forests as a legacy of historical land use change cannot meet the
increasing demand. This exemplifies the potential inconsistencies between model structure and forcing
data. In addition, under such a circumstance, old forests in $S_{age}$ simulation tend to have higher biomass
density than the ageless forests in $S_{ageless}$, because in $S_{age}$ these mature forests remain intact throughout the
whole simulation, while the ageless forests in $S_{ageless}$ are "degraded" due to all historical LUC activities.
This explains the slightly higher $E_{LUC\ harvest}$ in the $S_{age}$ simulation. This is also supported by the fact that
the difference in $E_{LUC\ harvest}$ between $S_{ageless}$ and $S_{age}$ by the S2b simulations is smaller than using the S2
simulations. This is because in S2b, emissions from wood harvest are quantified by including harvest on
top of net land use change only, thus applying harvest in both $S_{ageless}$ and $S_{age}$ to mature forests whose
biomass stocks have not been influenced by land turnover, so that $E_{LUC}$ from harvest in the end differs
little between $S_{ageless}$ and $S_{age}$.

**3.2 Spatial distribution of land use change emissions**
Figure 3 shows the spatial distribution of cumulative $E_{LUC}$ for 1501–2005 from different LUC processes
in the $S_{ageless}$ simulations (Fig. 3a–c), the difference in $E_{LUC}$ between $S_{age}$ and $S_{ageless}$ simulations (Fig. 3d–
f), corresponding net forest area change (Fig. 3g) and areas subject to land turnover (Fig. 3h) and wood
harvest (Fig. 3i). The spatial pattern of $E_{LUC\ net}$ generally resembles that of forest area loss, with large
areas of forests being cleared and corresponding high $E_{LUC\ net}$ in eastern North America, South America
and Africa, southern and eastern Asia, and in central Eurasia (Fig. 3a, Fig. 3g). Central and Eastern
Europe show some increases in forest area but carbon emissions from net land use change persists,
probably because forest recovery happened in recent times and carbon accumulation in recovering forests
is not yet big enough to compensate for historical loss (e.g., see Fig. 5f). Depending on different regions,
$E_{LUC\ net,\ age}$ is slightly higher (e.g., along the boreal forest belt in central Europe and Asia, woodland
savanna in South America) or lower (e.g., part of Africa and Australia) than $E_{LUC\ net,\ ageless}$ (Fig. 3d). This
difference between $S_{age}$ and $S_{ageless}$ is in general of rather low magnitude (<0.5 kg C m$^{-2}$ over 1501-2005).
It mainly depends on the age classes of forests to be cleared in the $S_{age}$ simulation and how the forest
biomass density compares with that from $S_{ageless}$ simulation and whether biomass density of the single
ageless mature patch is diluted or not with establishment of young forests.

In the LUH1 data set, shifting cultivation (land turnover here) is limited to the tropical region (Fig. 3h), as
in the original LUH1 forcing data. Tropical Africa is the region with most of the turnover activities, and
consequently has highest $E_{LUC\ turnover}$. Note the peripheral of Amazon basin also show active shifting
cultivations and resulting carbon emissions (Fig. 3b, Fig. 3h). $E_{LUC\ turnover,\ age}$ is in general lower than $E_{LUC}$
$_{tunrnover,\ ageless}$ everywhere except at the northern fringe of northern African woodland savanna (Fig. 3e).
Last, wood harvest mainly occurs in temperate and boreal forest in Northern Hemisphere (Europe and
central Siberia, eastern North America and southern and eastern Asia) and tropical forests including those
of Amazon forest, in central Africa and tropical Asia, with corresponding carbon emissions (Fig. 3c, Fig.
3i). $E_{LUC\ harvest,\ age}$ is a higher source than $E_{LUC\ harvest,\ ageless}$ for most of the harvested regions, which mainly
results from the model feature as explained above.

**3.3 Simulated regional LUC emissions**
Estimated carbon emissions since 1900 from different regions are shown in Fig. 4, with emissions from
each LUC source for $S_{ageless}$ simulation being shown in Fig. S8. The corresponding areas subject to the
three LUC processes with forests being mainly involved are shown in Fig. 5. We also compared our
estimations by Stocker et al. (2014), where the LUC emissions are simulated with a different vegetation
model (LPX-Bern) but contributions of each individual LUC process is quantified with a similar approach
as ours. Both studies are forced by the LUH1 data set, although actual areas undergoing different land use
change activities may slightly differ because of different LUC implementation strategies. As shown in
Fig. 5, in spite of incessant episodic forest gains, for most time in most regions, historical net forest
change was dominated by forest loss, except for the latter half of the 20$^{th}$ century in Western Europe,
Former Soviet Union (FSU), and for the time period after 1970 in Pacific Developed Region. Meanwhile,
land turnover and wood harvest persisted for most regions, although their magnitudes varied over time.
While forest gain can lead to carbon uptake, it could be outweighed by emissions from simultaneous
forest loss (note here both forest loss and gain occurred as a result of net land use change within the same
region but not within the same grid cell), land turnover and wood harvest. Thus it is not surprising that for
most regions and most time, LUC impacts on carbon cycle are diagnosed as emissions, except for the
latter half the 20[th] century for Former Soviet Union (Fig. 4).

The two estimations of LUC emissions from our study and Stocker et al. (2014) are in general agreement
for most of the regions, including their temporal variations. Emissions globally are dominated by Central
and South America and Africa & Middle East. Emissions increased in both regions since 1900, and a
peak of emissions occurred around the middle of the 20[th] century in Africa and around 1980 in Central
and South America (Fig. 4a, 5b). Emissions in Stocker et al. (2014) show similar temporal variations for
these two regions as in our study. The peak of emissions in Africa & Middle East around 1950 is clearly
dominated by a peak of forest loss due to net land use change (red line in Fig. 5b), and a surge of forest
loss due to land turnover that has accelerated between 1940 and 1960 (green line in Fig. 5b). After that
emissions decreased slightly, mainly due to the stabilized land turnover activities and a drop in area of net
land use change. Then the emissions slightly increased again around 1980s, due to an increase in forest
loss of net land use change (red line in Fig. 5b) and wood harvest (cyan line in Fig. 5b).  In contrast, even
with a similar peak of forest loss due to net land use change in Central and South America as in Africa &
Middle East, as shown in Fig. 5a (red line), emissions in the former region continued to increase until
1980s (Fig. 4a), mainly due to continuous growing of forest losses resulting from land turnover (green
line in Fig. 5a).

Both South & Southeast Asia and China Region showed steady increase in emissions up to c.a. 1990s
(Fig. 4c, 5d). In the former case, this is likely driven by continuous growing land turnover and wood
harvest; in the latter case, it is more likely driven by growing net forest loss (Fig. 5c, 6d). The peak in
emissions around 1990s in China Region echoes a peak in net forest loss (red line in Fig. 5d). Stocker et
al. (2014) shows slightly higher emissions than our estimates for South & Southeast Asia, and lower
magnitude in China Region, but with similar temporal patterns in both regions. For the three regions
where land turnover activities are included in the LUH1 data set (i.e., Central and South America, Africa
& Middle East and South & Southeast Asia), there are some periods during which $E_{LUC\ ageless}$ is clearly
higher than $E_{LUC\ age}$. These mainly correspond to the time when land turnover area either showed
decelerated growth or stabilized, being roughly after 1970 in Central and South America (Fig. 4a), 1965-
1985 in Africa & Middle East (Fig. 4b), and after 1980 in South & Southeast Asia (Fig. 4c).

North America shows most clearly the legacy impact of past LUC activities on LUC emissions. For the
period 1900–1940, carbon emissions in North America gradually decreased even though areas subject to
forest loss and wood harvest showed slight increases (Fig. 4e, Fig. 5e). This is likely due to the year 1900
is preceded by a peak of net forest loss, which yielded a high emission legacy for the several beginning
decades in the 20[th] century (data not shown). LUC emissions and sinks in Pacific Developed Region and
Europe are of very small magnitudes, despite a high forest wood harvest area in Europe. This is because
in general $E_{LUC\ harvest}$ is small compared to $E_{LUC\ net}$, probably due to the biomass accumulation in re-
growing forest (Fig. S8). The carbon sink brought about by net forest gain is the most prominent in
Former Soviet Union (blue line in Fig. 5h), where a peak of forest gain around 1950s lead to a sustained
sink of ~0.1 PgC yr$^{-1}$ for the latter half of the 20[th] century (Fig. 4h), however, concurrent sink is not seen
in Stocker et al. (2014).

**4 Discussion**
**4.1 Impacts on estimated $E_{LUC}$ by including gross land use change and sub-grid secondary forests**
The advancement in this study in comparison with previous works, as far as we know, is the explicit
inclusion of differently aged sub-grid secondary land cohorts in a DGVM. Although secondary lands have
been represented in some DGVMs in previous studies (Shevliakova et al., 2009; Stocker et al., 2014;
Yang et al., 2010), here we incorporated the concept of rotation cycle. This is particularly important in
simulating the carbon cycle impacts of gross land use change, such as wood harvest and shifting
cultivation that often have certain rotation cycles. Because secondary lands, especially young re-growing
forests, have lower biomass carbon stock than primary mature forests, related land use change emissions
tend to be lower than otherwise modeled without sub-grid age dynamics, which is equivalent to clearing
of mature forests before they're extensively disturbed. Our results using a fixed rotation length of 15
years in shifting cultivation in tropical regions demonstrate that by explicitly including secondary forest
cohorts, estimated carbon emissions for 1501–2005 are reduced from 45.4 Pg C to 27.4 Pg C, or 40%
lower with age dynamics than without.

Nonetheless, it should be noted that this conclusion is obtained using a constant 15-year rotation length in
shifting cultivation in the tropics, to be consistent with the rule of LUH1 for this LUC process. To test the
sensitivity of $E_{LUC\ turnover}$ to the rotation length in $S_{age}$ simulations, we performed further five alternative S2
simulations, all starting from 1861 based on the system state of 1860 obtained by the default S2
simulation, but with the primary target cohort in land turnover varying among the other five cohorts
except Cohort$_3$, which is the default target cohort. The results are presented in Fig. S9. $E_{LUC\ turnover}$ over
1861–2005 increases in a roughly linear way with the assumed woody mass of forest cohorts that are
cleared in shifting cultivation, with emissions increasing by 5.3 Pg C for each kg C m$^{-2}$ increase in cohort
woody mass. $E_{LUC\ turnover,\ ageless}$ is slightly higher than $E_{LUC\ turnover,\ age}$ when cohorts with ~15 years are
cleared primarily. Increasing rotation lengths thus leads to higher emissions than in $S_{ageless}$ simulations in
this case. This suggests the critical importance of the rotation length for land turnover, i.e. the residence
time of agriculture in shifting cultivation systems.

Table 3 summarized estimations of $E_{LUC}$ from different studies by including both net transitions and gross
land use change, and the contributions to total emissions by including gross transitions. All studies show
that including gross land use change increased estimated carbon emissions. Stocker et al. (2014) reported
that gross change contributed 15% to total emissions, whereas Wilkenskjeld et al. (2014) reported a much
higher contribution of 38%. Hansis et al. (2015) by using a bookkeeping model, reported a 22–24%
contribution from gross change if cleared lands are primarily from primary lands, in contrast to a small
contribution of only 2% if cleared lands are exclusively from secondary lands. For the $S_{ageless}$ simulation
in the current study, the contribution from gross land use change to total emissions is 20%, falling in
between Stocker et al. (2014) and others including the 28% contribution by gross change in the tropics
reported by Houghton (2010). However, the simulation by including secondary land (i.e., $S_{age}$) gives a
lower gross land use change contribution (15%) than $S_{ageless}$, although this result depends on the assumed
constant 15-year rotation length in shifting cultivation in the tropics. In general, the same model yields
lower contribution of gross changes by converting dominantly secondary land versus primary land (our
study and Hansis et al., 2015). Among different models/methods, the ones including secondary lands
(Houghton, 2010; Stocker et al., 2014) tends to yield lower contribution of gross changes than those do
not  (Wilkenskjeld et al., 2014). Although the exact percentage might differ depending on the amount of
gross changes included and the biomass stocks of the secondary lands being cleared, it seems that
contributions from gross land use change are lower when including sub-grid secondary lands.

We also expected $E_{LUC}$ from wood harvest to be smaller when including secondary forests, for the same
reason than shifting cultivation. However, we obtained a slightly higher $E_{LUC\ harvest,\ age}$ than $E_{LUC\ harvest,\ agelss}$,
mainly because there are not enough secondary forests available for harvesting in $S_{age}$, so that mature
forests with a higher biomass density than in $S_{ageless}$ are harvested according to the priority setting in the
model, which leads to higher emissions. This model feature was designed to address potential
inconsistencies between prescribed harvest area in the forcing data and (secondary) forest availability in
the model, to ensure that ultimately realized harvest area in the model is as close as possible to the
prescribed one. From the S2b simulations where wood harvest, instead of land turnover, is added on top
of net land use change, $E_{LUC\ harvest}$ derived from $S_{age}$ and $S_{ageless}$ are very similar because in both
simulations, forests with biomass close to the one of primary forests are harvested and their carbon stocks
are similar between $S_{age}$ and $S_{ageless}$. Finally, it should also be noted that reconstructions of forest wood
harvest are highly uncertain. For example, LUH1 data provides a total wood harvest amount of 102 Pg C
for 1850–2005 over forest and non-forest areas, whereas Houghton and Nassikas (2017) estimated as 130
Pg C. Our estimates of $E_{LUC\,harvest}$ using different approaches is 22.5–27.8 Pg for 1850–2005, close to the
estimated 25.3 Pg C for 1850–2015 by Houghton and Nassikas (2017).

In the current study, we implemented wood harvest based on input (LUC forcing) information on
harvested area rather than on wood volume or biomass. In the future, this process should be modified so
that harvested wood volume or biomass information is directly used in the model, to allow dynamic
decision on whether an old forest or secondary forest should be harvested. Using wood harvest volume or
biomass information would largely alleviate the uncertainty brought about by the unknown wood harvest
rotation length because the total amount of harvested biomass would be constrained (Houghton and
Nassikas, 2017).

We do not account for any LUC activities in the spin-up run and pristine ecosystems are assumed at the
beginning of the transient run in 1501. This set-up might cause a spike in emissions during the beginning
years in the transient simulation because ecosystem biomass stocks are high, due to a lack of historical
disturbance. Such a spike was evident in results by Stocker et al. (2014, blue and green lines in their Fig.
2) when land turnover is not accounted for during the spin-up in some of their simulations. The similar
model behaviour also presents in the results by Hansis et al. (2015, dark and light blue lines in their Fig.
4) using a bookkeeping model. In our study, a similar initial spike in $E_{LUC}$ shortly after 1501 is almost
invisible for the net land use change and land turnover (Fig. 2a–b), probably owing to very small
magnitudes of LUC area within the few years after 1501 (Fig. 2d–e). However, there is a clear peak in
$E_{LUC\,turnover}$ around 1520s (Fig. 2c), a likely impact of ignoring spin-up LUC process, given that a
significantly larger-than-zero harvest area is prescribed for this period (Fig. 2f). In general, the impacts of
not including LUC in the spin-up process seem to be small in our results. This issue impacts much less
the comparisons focusing on emissions starting from 1850 in Table 3.

**4.2 Impacts on estimated emissions by initial biomass stock**
As shown in Fig. 2 and Table 3, our estimations of historical LUC emissions from both $S_{ageless}$ and $S_{age}$
simulations are lower than other studies for most time of history (albeit close to Stocker et al. 2014 before
ca. 1860). We compared in Table S1 the cumulative $E_{LUC}$ for 1850-2005 by our studies and several
previous studies. Our estimates (147 Pg C for $E_{LUC\,age}$ and 158 Pg C for $E_{LUC\,ageless}$) are lower than the
lower bound of other estimates (171 Pg C by Stocker et al. 2014). Estimations of Hansis et al. (2015) and
Gasser and Ciais (2013) using Hurtt et al. (2011) data set give rather larger estimates than others, being
261 and 294 Pg C, respectively. The median value of all previous estimates cited in Table S1 yields 210
Pg C, still much higher than our estimates.

The lower estimates of $E_{LUC}$ in our study are likely linked with underestimated global biomass carbon
stock in ORCHIDEE-MICT V8.4.2. The global biomass carbon stock simulated by our model at 1500
prior to any land use change is 365 Pg C, and increases to 510 Pg C at 2005 in the S0 simulations (i.e.,
assuming no LUC activity). The simulated global biomass remains almost unchanged in the S3
simulations where all three LUC processes are included. Avitabile et al. (2016) merged two tropical
aboveground forest biomass data sets from Saatchi et al. (2011) and Baccini et al. (2012) with northern
hemisphere volumetric forest stock growth data from Santoro et al. (2015). Their estimated global forest
biomass for aboveground only is 505 Pg C. Our simulated contemporary global total biomass stock (i.e.,
from S3 simulations) is thus even lower than their estimate for aboveground biomass.

Li et al. (2017) has identified emergent linear relationship between cumulative $E_{LUC}$ for 1901–2012 and
initial biomass in 1901 among the nine DGVMs of the Trends in Net Land-Atmosphere Exchange
(TRENDY-v2) project (http://dgvm.ceh.ac.uk/node/9) (shown in Fig. S10). They further used these
relationships to obtain an observation-constrained $E_{LUC}$ (horizontal orange line in Fig. S10) for 1901–
2012 that are independent of DGVMs, by reconstructing an initial biomass carbon stock in 1901(vertical
green line in Fig. S10) based on contemporary satellite observations of global biomass distribution. As is
shown in Fig. S10, carbon stocks are indeed underestimated in our model for a few regions and the globe,
compared to the satellite-based reconstructions of 1901 biomass in (Li et al., 2017b). We derive a
biomass-corrected cumulative $E_{LUC}$ for each region and the globe for 1901-2005, by using the
relationships between cumulative $E_{LUC}$ and initial 1901 biomass among different DGVMs, as shown in
Fig. S10. The biomass-corrected cumulative $E_{LUC}$ for 1501–2005 and 1850–2005 are further derived, by
assuming the same ratio between biomass-corrected and original cumulative $E_{LUC}$ for these two periods
against that of 1901–2005. The original and biomass-corrected cumulative $E_{LUC}$ for 1501–2005, 1850–
2005 and 1901–2005 and the correction ratios for each region and the globe are summarized in Table S2.
However, these corrected values should be taken with caution and they're not fully quantitatively valid.

The biomass-corrected global cumulative $E_{LUC}$ for 1850–2005 are 174–207 Pg C for the $S_{ageless}$
simulation, and 161–194 Pg C for the $S_{age}$ simulation (Table S1), larger by 10–30% than the original
values. These are in closer agreement with the median value of previous studies (210 Pg C). In addition,
the magnitude of historical LUC activities actually included in our simulation is lower than that
prescribed in the original LUH1 data set, as an inevitable result from the reconciliation between LUH1
data set and the used ESA CCI 2005 PFT map (Fig. S2, see also Sect. 2.2). If these omitted transitions
had been taken into account, estimated cumulative $E_{LUC}$ for 1850–2005 would have reached 172–204 Pg
C for the $S_{age}$ simulation, and 191–226 Pg C for $S_{ageless}$ simulation, assuming that emissions increase
proportionally with the areas subject to land turnover transitions. However, the omitted net transitions
between natural land and agricultural land might not lead to substantial increase in $E_{LUC}$ considering our
historical loss of forest area over the globe largely matches that by Peng et al. (2017) whose forest loss is
further based on Houghton et al. (2003) and the FAO data (Fig. S3) and therefore the additional
conversion of natural lands to agriculture would come from mainly natural grasslands. If we further
account for the missing land turnover areas from spatial upscaling of the LUH1 data, then the estimated
$E_{LUC\ turnover}$ would be even higher.

**4.3 Land use and management processes in DGVMs in relation to forest demography**
Forest demography is an important factor in determining forest carbon dynamics at both stand and
regional scales (Amiro et al., 2010; Pan et al., 2011). Natural disturbances (such as fire, wind and insect)
and land use change including land management are two primary factors creating spatial heterogeneity in
forest age. As more and more forests are now under human management with differed intensities (Erb et
al., 2017; Luyssaert et al., 2014), sub-grid forest demography should be incorporated in DGVMs to
account for the management consequences. Furthermore, when making more accurate (and detailed)
account of regional carbon balances linked with land use change, other land cover types than forests
should be distinguished into different cohorts as well, because the presence of many nonlinear processes
(e.g., soil carbon decomposition) makes the simple averaging scheme as in the case where they're
represented with a single patch within the model a sub-optimal choice. This new model structure, to have
more than one cohort for the same land cover within a grid cell, as is partly explored by Shevliakova et al.
(2009) as well in a dynamic land model LM3V less complex than ORCHIDEE-MICT v8.4.2, will have
impact on simulated biogeochemical and biophysical processes, as partly demonstrated here.

However, despite these improvements in model structure, it remains a big challenge to "seamlessly"
integrate land use change forcing data into the model. The fundamental reason is that historical transitions
of land use change are not reconstructed in a way being internally consistent with DGVMs. The system to
build historical LUC transitions (so-called LUC model) and DGVMs may use different land cover types
so that conciliating the two land cover maps is inevitable. This will lead to loss of information in
incorporating forcing data into the model, as is pointed out also by Stocker et al. (2014). Second,
simulated forest biomass density might be different as well, so that the same amount of harvested wood
volume will translate into different forest areas in the LUC model and DGVMs. Recently progresses have
been made in DGVMs to represent forest stand structure and detailed management options (Naudts et al.,
2015), so that wood volume information can be used directly as a forcing in the model to drive forestry
decisions. Third, LUC model uses assumptions on rotation lengths of shifting cultivation or forest
management, and information generated there might not be consistent with forest age distribution in
DGVMs, as is the case in our study.

To overcome these obstacles and allow a more comprehensive integration of land use change information
into DGVMs, one possible route is to further develop DGVMs to partly embed functions of LUC models.
This will allow DGVMs to be used in an "inversed" manner than its current way of utilization. For
example, food demand could be used as an input, so that dynamical decisions could be made within the
model on how many croplands need to be created given the simulated crop yield by the crop module
inside the DGVM. The same case also applies on pasture. Grassland management modules within
DGVMs could generate information on meat and milk production etc., and this information could be used
to inverse the meat and mild demand into demanded pasture areas (Chang et al., 2016). Harvested wood
for a certain product usage might need wood with a specific diameter range, corresponding to a certain
forest age class given their simulated growth state, allowing the determination of both ages and areas of
forests to be harvested.

**5 Conclusions**
In this study, we investigated the impacts on estimated historical gross land use change emissions by
accounting for multiple sub-grid secondary land cohorts in a dynamic global vegetation model. The
model employed here is capable of representing the rotation processes in land use and land management
that mainly involve secondary forests, such as shifting cultivation and forest wood harvest.
Intermediately-aged secondary forests are given a high priority when forest clearing occurs in either
shifting cultivation or wood harvest, complemented by older forests if young ones are insufficient to meet
the prescribed land use transition. For the land use transition that entails a net change in the land cover,
clearing of forests start exclusively from mature forests and move sequentially to younger forests when
older ones are used up. This set of rules becomes indispensible when incorporating multiple sub-grid
secondary land cohorts and reconciling with external land use transition forcing data in the model. As
such, the simulated portfolio of secondary land cohorts within the model is driven by a reconstruction of
historical gross land use change.

Following the input data of land use transition reconstruction, we assumed a constant shifting cultivation
rotation length of 15 years in the tropics. We found that over 1501-2005, accounting for sub-grid
secondary land cohorts yields lower land use change emissions than not (176 versus 197 Pg C), which is
dominated by lower emissions from shifting cultivation (27 versus 46 Pg C or 40% lower in the former
case). This is because secondary forests with a lower biomass are allowed being cleared, instead of the
mature forests with a high biomass as in the approach to representing only mature forest in DGVMs. The
lower emissions from shifting cultivation when accounting for sub-grid multiple land cohorts highly
depend on the assumed rotation length. A set of sensitivity runs for Africa showed that a longer historical
shifting cultivation rotation length leads to higher associated emissions. This highlights the need for more
reliable reconstructions of the areas as well as the historical rotation lengths of shifting cultivation, and in
general of the land turnover process, to reduce uncertainty on $E_{LUC}$. Our results show that although gross
land use change as a previously neglected LUC emission component has been included by a growing
number of DGVMs, its contribution to overall $E_{LUC}$ remains uncertain and tends to be overestimated by
models ignoring sub-grid secondary forests.

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

**Data availability**
All data used to generate the figures are available in the Supplement of this paper.
**Competing interests**
The authors declare that they have no conflict of interest.

**Acknowledgments**
C.Y., P.C. and W.L. acknowledge support from the European Research Council through Synergy grant
ERC-2013-SyG-610028 "IMBALANCE-P".  W.L. and C.Y. are also supported by the European
Commission-funded project LUC4C (No. 603542). The authors thank the two reviewers for their
constructive comments that help to improve the manuscript quality.

**Tables and figures**
Table 1 Factorial simulations to quantify $E_{LUC}$ from each of the LUC processes considered: net land use
change ($E_{LUC\ net}$), land turnover ($E_{LUC\ turnover}$) and wood harvest ($E_{LUC\ harvest}$), with $E_{LUC\ all}$ being carbon
emissions from all the three processes. The plus sign ("+") indicate that the process in question is
included, with $S0_{ageless}$ ($S0_{age}$) having no LUC activities to $S3_{ageless}$ ($S3_{age}$) including all LUC processes.
$E_{LUC}$ is quantified as the difference in net biome production (NBP) between simulations without and with
LUC. To explore the uncertainties by using a fully additive approach, we included an alternative S2b
simulation, which includes net land use change and land turnover. $E_{LUC\ turnover}$ and $E_{LUC\ harvest}$ are
consequently calculated using this alternative simulation as well.

| Simulations and LUC processes included | | | |
|---|---|---|---|
| Simulations | Net land use change | Land turnover | Wood harvest |
| $S0_{ageless}$ ($S0_{age}$) | | | |
| $S1_{ageless}$ ($S1_{age}$) | + | | |
| $S2_{ageless}$ ($S2_{age}$) | + | + | |
| $S3_{ageless}$ ($S3_{age}$) | + | + | + |
| $S2b_{ageless}$ ($S2b_{age}$) | + | | + |
| **Calculation of $E_{LUC}$** | | | |
| No age dynamics ($S_{ageless}$) | | With age dynamics ($S_{age}$) | |
| $E_{LUC\ net,\ ageless} = NBP_{S0,\ ageless} - NBP_{S1,\ ageless}$ | | $E_{LUC\ net,\ age} = NBP_{S0,\ age} - NBP_{S1,\ age}$ | |
| $E_{LUC\ turnover,\ ageless} = NBP_{S1,\ ageless} - NBP_{S2,\ ageless}$ | | $E_{LUC\ turnover,\ age} = NBP_{S1,\ age} - NBP_{S2,\ age}$ | |
| $E_{LUC\ harvest,\ ageless} = NBP_{S2,\ ageless} - NBP_{S3,\ ageless}$ | | $E_{LUC\ harvest,\ age} = NBP_{S2,\ age} - NBP_{S3,\ age}$ | |
| $E_{LUC\ turnover,\ ageless\ S2b} = NBP_{S2b,\ ageless} - NBP_{S3,\ ageless}$ | | $^{*}E_{LUC\ turnover,\ age\ S2b} = NBP_{S2b,\ age} - NBP_{S3,\ age}$ | |
| $E_{LUC\ harvest,\ ageless\ S2b} = NBP_{S1,\ ageless} - NBP_{S2b,\ ageless}$ | | $^{*}E_{LUC\ harvest,\ age\ S2b} = NBP_{S1,\ age} - NBP_{S2b,\ age}$ | |
| $E_{LUC\ all,\ ageless} = NBP_{S0,\ ageless} - NBP_{S3,\ ageless}$ | | $E_{LUC\ all,\ age} = NBP_{S0,\ age} - NBP_{S3,\ age}$ | |


Table 2 Determination of woody biomass thresholds for different age classes of forest PFTs. The
thresholds of woody biomass are determined by looking up via the biomass-age curve (Eq. 2), the ratio of
woody biomass to the maximum biomass ($B_{max}$) that correspond to certain ages (years), followed by
multiplying this ratio with equilibrium biomass ($B_{max}$) at each grid cell. Numbers in the table indicate the
ratio of woody biomass to the maximum woody biomass ($B_{max}$ in Eq. 2), and the numbers in parentheses
indicate the corresponding forest age.

| Forest cohorts | Tropical forest | Temperate forest | Boreal forest |
|---|---|---|---|
| Age1 | 0.1 (3 year) | 0.07 (3 year) | 0.04 (3 year) |
| Age2 | 0.26 (9 year) | 0.22 (10 year) | 0.19 (15 year) |
| Age3 | 0.39 (15 year) | 0.40 (20 year) | 0.34 (30 year) |
| Age4 | 0.6 (27 year) | 0.6 (35 year) | 0.6 (65 year) |
| Age5 | 0.8 (48 year) | 0.8 (64 year) | 0.8 (114 year) |
| Age6 | 1.2 (>48 year) | 1.2 (>64 year) | 1.2 (>114 year) |


Table 3 LUC emissions for 1501–2005 (Pg C) from different processes quantified by different approaches
(see Table 1 for detailed calculations of various $E_{LUC}$).

| | No age dynamics | With age dynamics | Emission change in $S_{age}$ relative to $S_{ageless}$ (%) |
|---|---|---|---|
| $E_{LUC\ net}$ | 123.7 | 118.0 | -4.6% |
| $E_{LUC\ turnover}$ | 45.4 | 27.3 | -40% |
| $E_{LUC\ turnover\ S2b}$ | 39.9 | 25.1 | -37% |
| $E_{LUC\ harvest}$ | 27.4 | 30.8 | 12% |
| $E_{LUC\ harvest\ S2b}$ | 32.9 | 33.0 | 0.0% |
| $E_{LUC\ total}$ | 196.5 | 176.1 | 10% |


Table 4 Carbon emissions from gross and net land use transitions, contributions of gross transitions to the
total emissions from different studies, adapted from Hansis et al. (2015).

| Reference | Time period | E$_{LUC}$ (Pg C) | | Contribution of gross transitions, Pg C (%) |
|---|---|---|---|---|
| | | Gross transitions | Net transitions | |
| This study (With age dynamics) | 1850-2005 | 147 | 99 | 22(15%) |
| This study (No age dynamics) | 1850-2005 | 158 | 104 | 31(20%) |
| Hansis et al (2015) | 1500–2012 | 382 | secondary land only 374 | 8.5 (2%) |
| Hansis et al (2015) | 1500–2012 | 382 | primary land first 290 | 92.4 (24%) |
| Hansis et al (2015) | 1500–2012 | 382 | primary land last 296 | 85.8 (22%) |
| Stocker et al (2014) | 1850–2004 | 171 | 146 | 25 (15%) |
| Wilkenskjeld et al (2014) | 1850–2005 | 225 | 140 | 85 (38%) |
| Houghton (2010) | 1850–2005 | 156 | | (28%, tropics) |

[a]The last column gives the difference between the net LULCC flux estimates for gross and net transitions (absolute in Pg C and relative to the net LULCC flux for gross transitions).


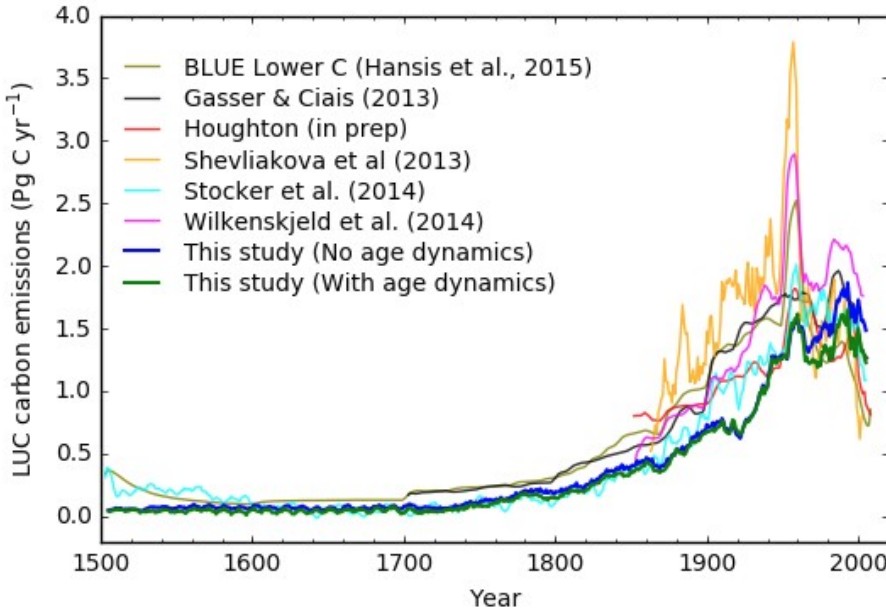


Fig. 1 Annual carbon emissions from historical land use change over the globe by our studies and from
other previous studies. Results of this study are smoothed using a ten-year average moving window; data
of other studies are from Figure 5 Hansis et al (2015) and are smoothed using a five-year moving average
window.

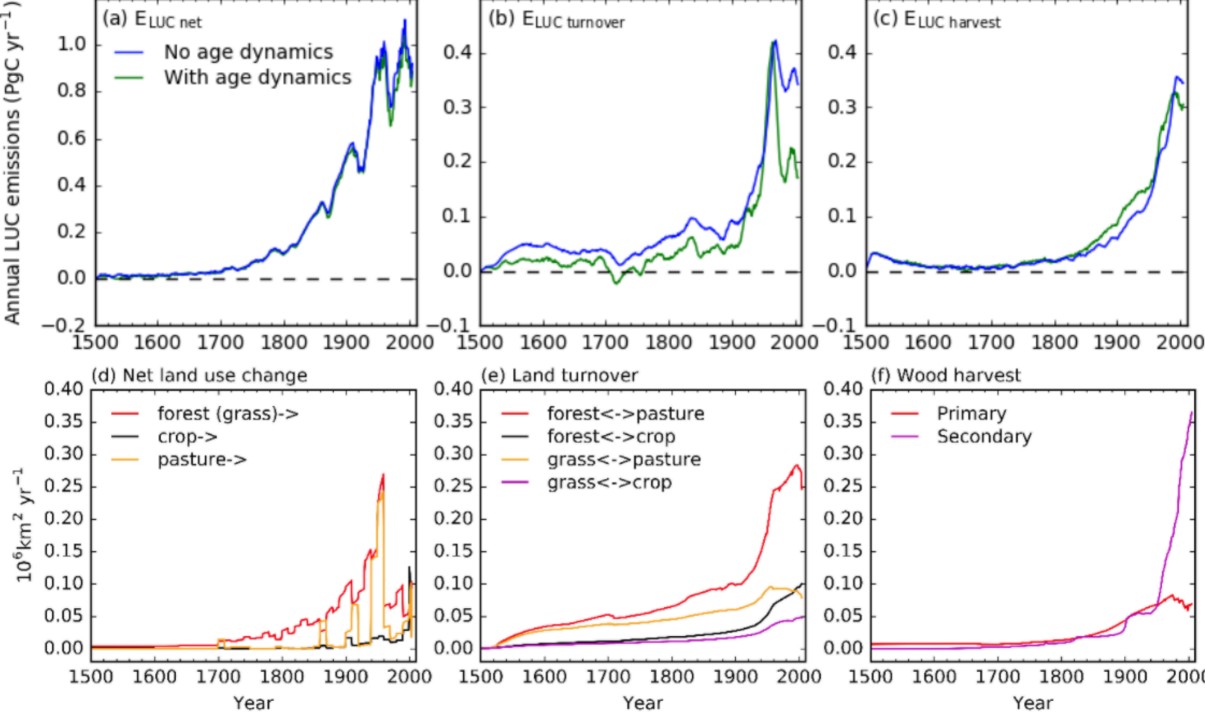


Fig. 2 Upper panels: annual carbon emissions since 1501 from different LUC processes, (a) net land use
change, (b) land turnover and (c) wood harvest. Data are smoothed using a ten-year average moving
window. Lower panels: annual time series of areas impacted by different LUC processes. (d) Area losses
of forest, grassland, cropland and pasture as a result of net land use change. Note that we assume equal
contributions by forest and grassland to agricultural land when backcasting historical land cover maps and
net land use transitions, thus area losses of forest and grassland are identical. (e) Areas subject to land
turnover. (f) Areas of wood harvest from primary and secondary forests.


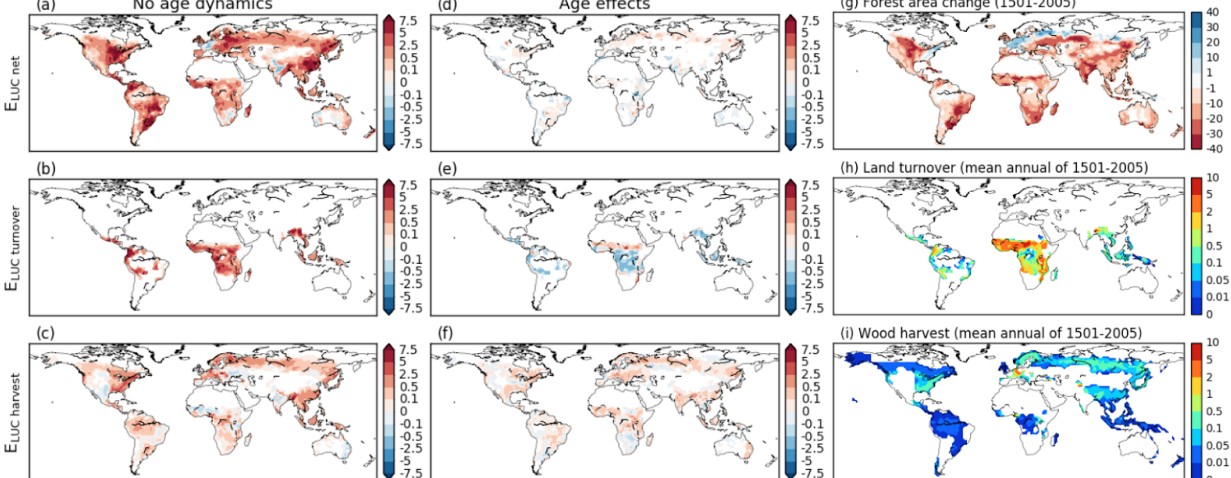


Fig. 3 (a)-(c): Spatial distribution of $E_{LUC}$ from different LUC processes by the simulation without sub-
grid age dynamics for 1501-2005 in unit of kg C m$^{-2}$, for (a) net land use change, (b) land turnover and (c)
wood harvest. Subplots (d)–(e) show the age effect as the difference between $E_{LUC\ age}$ and $E_{LUC\ ageless}$ for
each LUC process, with positive (negative) values indicating higher (lower) $E_{LUC}$ by the $S_{age}$ simulation.
(g) Cumulative forest loss as a result of net land use change for 1501–2005 as a percentage of grid cell
area. (h) Mean annual grid cell percentage impacted by land turnover over1501–2005. (i) Mean annual
grid cell percentage impacted by wood harvest (i.e., sum of wood harvest on primary and secondary
forests) over 1501–2005.


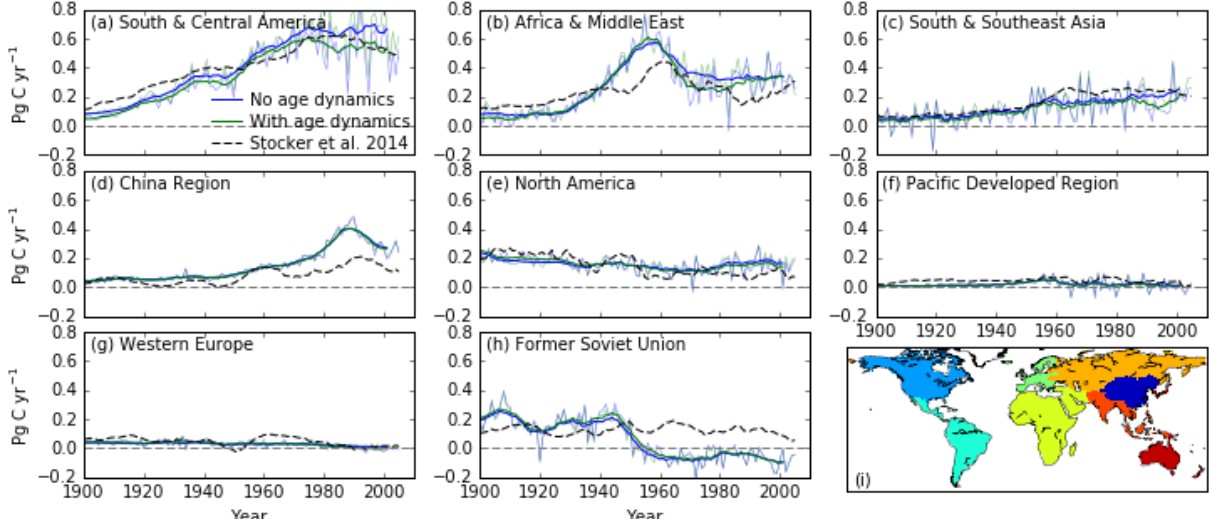


Fig. 4 (a)-(h) Temporal patterns of regional land use change emissions in comparison with those from
Stocker et al. (2014). Thicker solid lines indicate smoothed annual emissions by ten-year moving average
from our study, with blue (green) showing emissions from $S_{ageless}$ ($S_{age}$) simulations. Thinner solid lines
indicate unsmoothed annual emissions from our study. Gray dashed lines indicate estimations from
Stocker et al. (2014), smoothed by ten-year moving average. Regional segregation of the globe is shown
in the subplot (i).



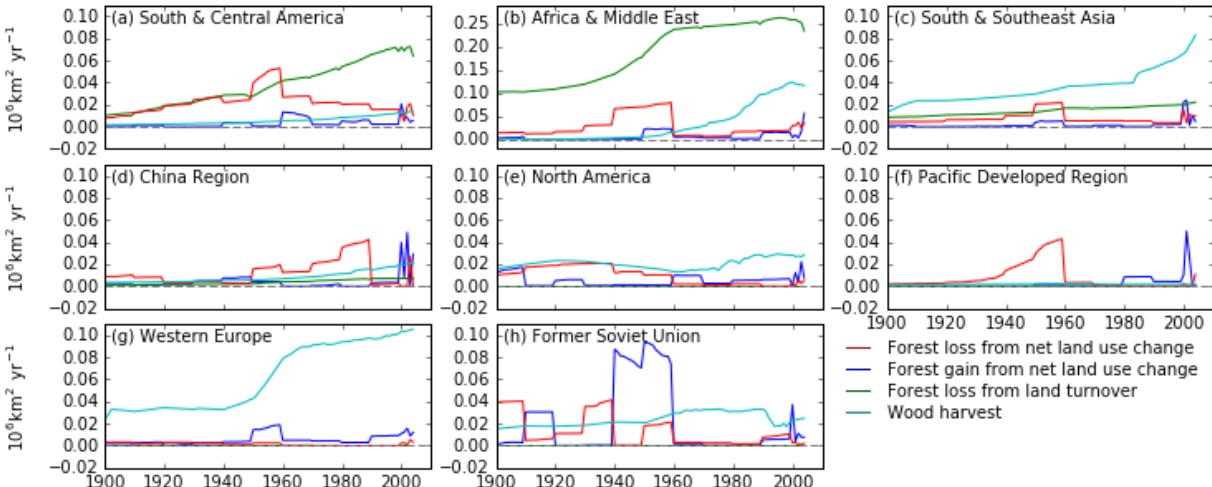


Fig. 5 Annual regional areas subject to land use change. Only land use change activities involving forests
are assumed to have dominant impacts on $E_{LUC}$ and are thus shown here: forest loss (red line) and gain
(blue line) from net land use change, occurring within the same region but not in the same model grid
cell; forest in involved in land turnover (green line) and wood harvest (cyan line), where forested land

1043 remain a forest after land use change. Note that the scale of y-axis is the subplot (b) is different from the

1044 others. See Fig. 4 for the spatial extents of different regions.