# Peer review of "Smaller global and regional carbon emissions from gross land use change when considering sub-2 grid secondary land cohorts in a global dynamic vegetation model 3 4 Chao Yue, Philippe Ciais, Wei Li 5 6 Laboratoire des Sciences du Climat et de l'Environnement, LSCE/IPSL, CEA-CNRS-UVSQ, Université"

_Biogeosciences, 2017_

## Referee Comment (RC1) · B. Stocker (Referee) · 19 Sep 2017

The present paper presents an application of the model described in Yue et al. (2017), GMDD, for global simulations covering the period where land use change (LUC) forcing data is available (1501-2005). Simulated cumulative emissions are 118 PgC for net land use plus 27.4 PgC for effects of sub-gridscale bi-directional land turnover (shifting cultivation type agriculture) plus 30.8 PgC for effects of wood harvesting. This amounts to a total of 176 PgC. This is at the lower end of the range of available estimates.

A special focus is put on the value of distinguishing age cohorts of land patches that have been affected by land conversion at different times in the past. The paper shows that not accounting for this effect increases estimates for cumulative LUC emissions. Authors explain that this is due to the generally higher average biomass density of converted land in simulations where no age cohorts are simulated.

Since effects of land turnover (shifting cultivation) and wood harvesting have been introduced into vegetation models, it has remained unclear what effect a distinction of age cohorts would have on simulated land use change emissions. The present paper addresses this knowledge gap and presents results from two simulations - one with age cohorts distinguished (Sage) and one without (Sageless). The reduction of the land turnover component of total emissions when comparing the two is extremes (Sage vs. Sageless) is 40

This is a notable contribution to the existing literature. However, its presentation and discussion in the context of the available literature is unsatifying and some parts misleading. Moreover, the present paper has substantial overlap with Yue et al. (2017), currently under review in GMDD. These aspects should carefully be adressed in the next revision round. Below I'm listing these two major points and a few (a bit more) minor ones.

**Major**

The point that the presentation and discussion of results in the context of the available literature is unsatifying echoes critique raised in the reviews of Yue et al. (2017), available through https://www.geosci-model-dev-discuss.net/gmd-2017-118/discussion, in particular the comment by J. Nabel. The same applies to the present paper. Particular attention should be paid to discuss results in the face of findings by Arneth et al. (2017) and to accurately describe which of the previously published models account for age cohorts within non-agricutural land and how many cohorts are distinguished. An overview table would help. Au-
thors describe the  $S_{age}$  simulation as reflecting the "traditional approach" (I.181), implying that the age cohort distinction is itself a novelty. However, it is not. Already Shevliakova et al. (2009) distinguished multiple cohorts. Stocker et al. (2014) distinguished two cohorts (primary and secondary land). Only the model described in Reick et al. (2013) and applied by Wilkenskjeld et al. (2014) makes no distinction between age cohorts. The LPJ-GUESS model (Smith et al., 2014) explicitly tracks C pools of land patches (cohorts) subjected to stochastic disturbance.  $S_{ageless}$  thus reflects an arguably extreme case and is not reflective of any "traditional approach". Having said that, an improved introduction and discussion will address this concern.

 My second major concern concerns the overlap with Yue et al. 2017, where the model applied here is described more extensively. Although authors only refer to their "idealized site-scale simulations" presented in Yue et al. (2017), it should be noted that also regional scale simulations, covering southern Africa, are presented therein and the main conclusion of that paper is identical to the main conclusion of the present paper - namely that accounting for age cohorts reduces the land turnover effect contribution to total LUC emissions. I raised this issue also as a reviewer for the GMDD paper and wrote:

The present paper [GMDD] was submitted on 14 May 2017. On 26 July 2017, Yue, Ciais and Li submitted a paper to Biogeosciences Discussions (https://www.biogeosciences-discuss.net/bg-2017-329/), where the same model is applied to investigate essentially the same questions, but this time at the global scale. The regional focus of the present paper on southern Africa may appear arbitrary at first, but makes sense. Apparently, authors preferred to devote a full paper to model description and evaluation and a second full paper to a global application. In my view, this is a viable way to go and the large work that went into developing this model warrants two separate papers. However, I find the delineation of their respective scope a bit unsatisfying. Readers will likely be left ask-
ing themselves why authors didn't present results from global simulations in the present (GMDD) paper - a relatively small additional step in terms of additional work. Simultaneously, readers of the BGD paper might be left wondering what the additional insight of that paper is after already the GMDD paper concluded that accounting for separate age cohorts reduces the effect of gross versus net LUC emissions.

The same issue applies vice-versa, i.e. to the present (BGD) paper. I further suggested to reinforce the value of the GMDD paper in terms of its model documentation and dissemination aspects. The present paper could for example gain in its value if the age-cohort effect is investigated not only for the two extremes (1 and 6 cohorts) but for additional numbers of cohorts, to establish a functional relationship between the number of cohorts and emissions. This would address also my previous point and would allow for a better comparison with models that distinguish between primary and secondary land (2 cohorts). Of course, this is just a suggestion, but I do encourage that the authors find a solution to finding a better delineation between their parallel submissions currently under review here and in GMDD.

**Minor**

Results of (residual) land sink (I.324-331) are confusing if not misleading. Authors find 89.2 PgC for 1959-2005 and compare this to the residual land sink from the global carbon budget (Le Quere et al., 2016). This addresses the question whether ORCHIDEE can simulate the land C sink as a result of changing environmental conditions, not anthropogenic LUC. This is a different question and out of scope for the present article. I suggest the paragraph I.324-331 to be dropped. Implications of higher LUC emissions simulated by models accounting for gross land use transitions as opposed to models simulating only net land use change are discussed by Arneth et al., 2017, where ORCHIDEE participated as
well. This point should not repeated here.

- It should be discussed that decisions with respect to priority of forest age cohorts used for conversion are unknown at the global scale.
- "Age classes for forest PFTs are distinguished in erms of woody biomass, while those for herbaceous PFTs are defined using soil carbon stock" (I.156): Discuss wether this definition is a problem when biomass and soil C stocks change in response to environmental conditions. I guess the simulated age distribution is therefore not an interpretable modelled quantity.
- "the land turnover resulting from the upscaling of 0.5° to 2° is not included" (I.240). This can be quite substantial. When transition maps are aggregated to a lower resolution for each transition separately, then this additional land turnover should be automatically included. How come it is not?
- "Following LUH1 (Hurtt et al., 2011), we assume that no land use change occurs during the model spin-up." (I.249). See my comment in the reviews of Yue et al. (2017), available through https://www.geosci-model-dev-discuss.net/gmd-2017-118/discussion, regarding model spin up:

Fig. 6 [in the GMDD paper] shows that if a constant land turnover rate is applied during the transient simulation, but not during spinup, biomass C stocks attain the "wrong" equilibrium. I.e. stocks decline after being subjected to continuous land turnover to a new steady state, reached after around 50 years (under a tropical climate). Soil C stocks likely take longer to attain a new steady state and in cold climates even more so. If simulations are evaluated from the start of the transient simulation, then land-atmosphere C fluxes related to reaching this new steady state confound results. How is this treated when, for example, doing a historical simulation starting in 1850? Shouldn't a continuous land turnover pattern be applied already during spin up in order to avoid these disequilibrium fluxes?

BGD
- Eq. 1 (I.256): Why is this decomposition defined here but no results for separated components are shown. Is Eq. 1 really necessary?
- I.363-375: It's important to note that harvest data used here specifies the harvested forest area. LUH alternatively provides harvested wood mass as a forcing dataset. Results presented here are subject to this choice and to the predefined priority rules (which age cohort to harvest first). According to I.172, the same priority rules are specified for land turnover and wood harvest, that is, middle-aged forest is harvested with a priority. Is this plausible? It may at least be equally plausible to assume that the oldest patch is harvested first as it has the highest biomass. In that case, the Sage simulation should have higher wood harvest-related emissions and the difference to Sageless should be small.
- I.542-543: Mention here how these compare to the un-corrected values.
- I.611: What does "down-estimate" mean?
- I. 615 (Conclusions): "This [accounting for cohorts] will lead to a lower-thanassumed so-called residual land CO2 sink on undisturbed land, which is inferred from the net balance of emissions from fossil fuel and land use change, and CO2 sinks in the atmosphere and ocean". This is a change of a change (age cohort effects on top of gross vs. net land use change effect) and the conclusion for a lower than expected residual land sink might appear confusing after Arneth et al. (2017) concluded a likely higher-than-expected residual land sink.

BGD

---

## Referee Comment (RC2) · S. Wilkenskjeld (Referee) · 27 Sep 2017

Yue and co-authors do in this paper demonstrate how inclusion of differently aged forests in the ORCHIDEE DGVM leads to reduced global carbon emissions (CE) from land use changes (LUC) during the period 1501-2005. This reduction is mainly attributed to the part of the CE which stems from shifting cultivation in the tropics (which they also included as a new feature in ORCHIDEE)). The authors systematically quantify the contribution of different processes (net LUC, shifting cultivation and wood harvest) to the total CE from LUC (ELUC). The study is thus an important contribution to

quantifying the ELUC which clearly demonstrates the importance of the inclusion of many aspects of vegetation dynamics and LUC to obtain accurate estimates of ELUC.

The paper is in general clearly written (though the authors at some places tend to repeat themselves), well structured and easy to read.

The main part of the description of the model development has been put in an accompanying paper "Representing anthropogenic gross land use change, wood harvest and forest age dynamics in a global vegetation model ORCHIDEE-MICT (r4259)", Global Model Development Discussions, 2017-118 (hereinafter GMD118), where the model functionality is demonstrated in an idealized site study and a regional study in South Africa. Since these two papers are closely related, some of my comments (including the main comment on the setup on the S-experiments) below also apply to GMD118 (unfortunately I missed the discussion deadline for GMD118).

The idea to separate the work in a development and an application part seems nice, but the separation between the two papers is not very clear: A lot of the model description is repeated in the present paper, and the analysis methods and results are very similar for the "South Africa" study and the global study which suggest to replace the results of the "South Africa" study with those of the global one in GMD118.

Though the papers (present and GMD118) represent a valuable contribution to the quantification of ELUC and its originating processes, there are a number of issues to be addressed at different level of severity:

**Major:**

Though qualitatively the major conclusion of the paper (effect of introducing age classes on gross transitions LUC) is obvious, unfortunately the experimental setup is not optimal for supporting this conclusion quantitatively. The authors use an "additional process approach" by starting with a model without any LUC (their S0), then adding net transitions (S1), gross transitions (also called "turnover", S2) and finally wood harvest

(S3). Such an approach only delivers a best guess for the last step - i.e. the wood harvest. However the main conclusion is about the turnover and the result does thus ignore the differences in the effects of wood harvest between the different experiments, which are clearly present (e.g. their increase in ELUC\_harvest from ageless to age). To provide a best guess on the effect of turnover, an additional experiment (I call it S4), including net transitions and wood harvest but ignoring turnover, would be needed. The turnover effects are then calculated from the difference between S4 and S3 instead of between S2 and S1. This could either be used to throw out S2 (the S2 setup is - to my knowledge - not used by any model, and thus is only usable to provide good estimates on the effects of wood harvest, not for model intercomparison) or to turn the general experimental structure into a "subtractive process approach", based on the "best guess" experiment (S3) and analyzing the effects of the different processes by removing them individually (turnover by comparing to S4, harvest by comparing to S2). In the first case the quality of the ELUC from wood harvest will be degraded, in the latter, some structural changes are needed to the paper.

I don't see the added value of the "South Africa" study in GMD118 in addition to the idealized site level study (also in GMD118) and the global study presented in this paper. The description of the "South Africa" sub-study in GMD118 is very short and hardly complete (e.g. which initial vegetation distribution was used?, were the LUH1 data backcast as in the present global study?)

A reasoning and discussion of the validity and influence of the priority rules for turnover and wood harvest is absent in this paper though some discussion is included in GMD118. This needs to be added or at least referenced and could also advantageously be extended.

The authors several times mention "inconsistencies between LUH1 and ESA-CCI-LC", but these problems may as well - at least in parts - stem from the choice of priority rules and the assumptions by Hurtt et al. (2011) for creating the global LUH1 data set. At least some comments attempting to disentangleing these effects should be made.

СЗ

See e.g. the discussion in Arneth et al. (2017) and references therein.

I. 507-543: Upscaling the ELUC based on scaling the total carbon to the TRENDY intermodel mean is very speculative and does - though it seems so - not add any quantitative information - specially not since the main focus of the paper is on the effects of including (or excluding) certain processes and not on the absolute ELUC numbers. I suggest to put the entire paragraph together to (essentially, not literally): "We have low absolute ELUC, relating to a low absolute carbon stock. These two quantities seems to be linearly related (Li et al. 2017)". This let the readers do the upscaling themselves being aware that this extrapolation is only qualitatively valid. This leaves Fig. S8, Table S2 and perhaps Table S1 (the main message can also be extracted from Table 3) obsolete.

The presentation let the model development seem entirely new, though Reich et al. (2013) contains a similar introduction of gross transitions and Shevliakova et al. (2009) introduced both vegetation with different age and gross transitions. These two studies must be taken into account in the description of the model development.

Minor:

Are S0-S2 and the Spinup entirely without wood harvest or do they use a fixed preindustrial (1500) wood harvest? If no harvest has been used, S3 will be subject to a "carbon chock" at the beginning of the transient run stemming from starting from a wrong equilibrium state and the absolute ELUC numbers - specially from S3 - are likely overestimated (S0 contains too much carbon).

Figure 6 needs to be introduced in paragraph 2.2 (likely with a lower number), since it actually do not show the results of the work of the authors but is rather a part of the description of the LUH1 data set. The figure is, however, absolutely necessary for the understanding of the results.

The numbers in Line 544-551 should also be introduced when introducing the LUH1

data set (paragraph 2.2). It is rather important for evaluating the results to know that substantial fractions of some of the transitions in the LUH1 data set are ignored.

Was "apparent gross transitions" arising from the aggregation of LUH1 (which only contains gross transitions in the tropics) over multiple grid cells actively suppressed outside the tropics? If yes: Why? This seems to be an unnecessary loss of information.

The division of herbaceous vegetation into two age cohorts based on the soil carbon (SOC) is either insufficiently explained or only representative for a certain type of LUC. In line 53-54 of GMD118 the authors state: "SOC decreases when a forest is converted to cropland; SOC increases when a cropland is converted to pasture" indicating that young herbaceous vegetation can have SOC both higher and lower SOC than the previous vegetation. Furthermore it seems that the division ignores that the main part of the changes in SOC do not take place instantaneously at the time of LUC.

Technical:

It should be made clear earlier in the paper that the terms "shifting cultivation" and "turnover" are used interchangeably.

Please repeat the main quantitative findings of the study in the conclusions.

In some cases letters are swapped in the subscripts.

Figs. 4-6 and S7: Please swap the order of the sub-panels from column-wise to rowwise. This is used in Fig. 2 and is much more intuitive.

Figs. 5, 6 and S7: The order of the geographical regions seems totally random. Please introduce some "around-the-globe"-ordering as in e.g. v.d.Werf et al. (2010). I am not saying, that the authors should adopt the regions from v.d.Werf - just the systematic ordering principle.

Figs. 3d-3f, 6, S3, S4 and S6: The unit Mkm2 is not a valid SI unit (double prefix). Please use "Mill. km2", "106 km2", "1012 m2" or rescale to e.g. "MHa" (which

would fit the numbers in Figs. 3 and 6 quite well).

Fig. 5 vs. 6: It is confusing that Fig. 5 starts in 1900 which Fig. 6 starts in 1800. The only thing mentioned in the paper before 1900 is - as far as I see - the peaks in North America. Does that need to be displayed?

Table 2: The main point of this table is the threshold fractions of Bmax used - the ages used for the determination are only relevant for the development stage and thus these are the numbers which should show up in brackets. Please either leave out "x Bmax" (described in the table caption) or add it everywhere - the mixture leaves the table rather confusing. The PFT-numbers are only of model internal relevance and should be removed.

In GMD118 I.477 and I.688 the LUH1 data set seems attributed to Hurtt et al. (2006) while the actual description of the data are in Hurtt et al. (2011).

The initial nomenclature is in my opinion more confusing (through unnecessary abstraction of rather simple expressions) than helpful and could be removed.

My personal opinion is that supplemental material should be kept at a minimum. For this paper this implies that the description of the backcast of the LUH1 data should rather be an appendix to the paper - or to GMD118 if the method was also applied here. Raw figure data should rather be "available upon request" than put in the supplement.

---

## Author Comment (AC1) · 26 Nov 2017

The present paper presents an application of the model described in Yue et al. (2017), GMDD, for global simulations covering the period where land use change (LUC) forcing data is available (1501-2005). Simulated cumulative emissions are 118 PgC for net land use plus 27.4 PgC for effects of sub-gridscale bi-directional land turnover (shifting cultivation type agriculture) plus 30.8 PgC for effects of wood harvesting. This amounts to a total of 176 PgC. This is at the lower end of the range of available estimates.

A special focus is put on the value of distinguishing age cohorts of land patches that have been affected by land conversion at different times in the past. The paper shows that not accounting for this effect increases estimates for cumulative LUC emissions. Authors explain that this is due to the generally higher average biomass density of converted land in simulations where no age cohorts are simulated.

Since effects of land turnover (shifting cultivation) and wood harvesting have been introduced into vegetation models, it has remained unclear what effect a distinction of age cohorts would have on simulated land use change emissions. The present paper addresses this knowledge gap and presents results from two simulations - one with age cohorts distinguished ($S_{age}$) and one without ($S_{ageless}$). The reduction of the land turnover component of total emissions when comparing the two is extremes ($S_{age}$ vs. $S_{ageless}$) is 40.

This is a notable contribution to the existing literature. However, its presentation and discussion in the context of the available literature is unsatifying and some parts misleading. Moreover, the present paper has substantial overlap with Yue et al. (2017), currently under review in GMDD. These aspects should carefully be adressed in the next revision round. Below I'm listing these two major points and a few (a bit more) minor ones.

[R1] We thank the reviewer for the general positive comments and the efforts to review both papers. Please see our point-to-point responses as below. Major revised texts are tracked in the updated manuscript.

**Major**

- The point that the presentation and discussion of results in the context of the available literature is unsatifying echoes critique raised in the reviews of Yue et al. (2017), availble through https://www.geosci-model-dev-discuss.net/gmd- 2017-118/discussion, in particular the comment

by J. Nabel. The same applies to the present paper. Particular attention should be paid to discuss results in the face of findings by Arneth et al. (2017) and to accurately describe which of the previously published models account for age cohorts within non-agricultural land and how many cohorts are distinguished. An overview table would help. Authors describe the $S_{age}$ simulation as reflecting the "traditional approach" (l.181), implying that the age cohort distinction is itself a novelty. However, it is not. Already Shevliakova et al. (2009) distinguished multiple cohorts. Stocker et al. (2014) distinguished two cohorts (primary and secondary land). Only the model described in Reick et al. (2013) and applied by Wilkenskjeld et al. (2014) makes no distinction between age cohorts. The LPJ-GUESS model (Smith et al., 2014) explicitly tracks C pools of land patches (cohorts) subjected to stochastic disturbance. $S_{ageless}$ thus reflects an arguably extreme case and is not reflective of any "traditional approach". Having said that, an improved introduction and discussion will address this concern.

[R2] We thank the reviewer for pointing to these works and this greatly helps to expand the discussion scope of our work. The introduction and discussion sections will be revised to take account into these works. In response to the reviewer's request, an overview table on current implementations of gross land use change in DGVMs is provided in the revised GMD manuscript as we think it's more appropriate there. The overview table will be cited in the revised BG manuscript. We also invite the editor and interested readers to check the discussions of the gmd-2017-118 paper as the reviewer's comments are highly related in these two papers, so are our responses.

- My second major concern concerns the overlap with Yue et al. 2017, where the model applied here is described more extensively. Although authors only refer to their "idealized site-scale simulations" presented in Yue et al. (2017), it should be noted that also regional scale simulations, covering southern Africa, are presented therein and the main conclusion of that paper is identical to the main conclusion of the present paper - namely that accounting for age cohorts reduces the land turnover effect contribution to total LUC emissions. I raised this issue also as a reviewer for the GMDD paper and wrote:

*The present paper [GMDD] was submitted on 14 May 2017. On 26 July 2017, Yue, Ciais and Li submitted a paper to Biogeosciences Discussions (https://www.biogeosciences-discuss.net/bg-2017-329/), where the same model is applied to investigate essentially the same questions, but this time at the global scale. The regional focus of the present paper on southern Africa may appear arbitrary at first, but makes sense. Apparently, authors preferred to devote a full paper to model description and evaluation and a second full paper to a global ap- plication. In my view, this is a viable way to go and the large work that went into developing this model warrants two separate papers. However, I find the delineation of their respective scope a bit unsatisfying. Readers will likely be left asking themselves why authors didn't present results from global simulations in the present (GMDD) paper - a relatively small additional step in terms of additional work. Simultaneously, readers of the BGD paper might be left wondering what the additional insight of that paper is after already the GMDD paper concluded that accounting for separate age cohorts reduces the effect of gross versus net LUC emissions.*

The same issue applies vice-versa, i.e. to the present (BGD) paper. I further suggested to reinforce the value of the GMDD paper in terms of its model documentation and dissemination aspects. The present paper could for example gain in its value if the age-cohort effect is

investigated not only for the two extremes (1 and 6 cohorts) but for additional numbers of cohorts, to establish a functional relationship between the number of cohorts and emissions. This would address also my previous point and would allow for a better comparison with models that distinguish between primary and secondary land (2 cohorts). Of course, this is just a suggestion, but I do encourage that the authors find a solution to finding a better delineation between their parallel submissions currently under review here and in GMDD.

[R3] In view of the reviewer's comments here, and the comments on our parallel gmd-2017-118 paper, we revised both papers to make a clearer delineation in their scopes: (1) Scopes are clearly defined in the introduction of each paper. The gmd-2017-118 paper focuses on model documentation and examination / illustration of model behaviour; the current paper focuses on model application on a global scale since 1850 and comparing simulated land use change emissions with other studies. (2) The figure on the carbon fluxes for Southern Africa in gmd-2017-118 has been removed. Only the Fig. 9 is kept there to illustrate the cohort dynamics with land use change in view of the hierarchical decision rules regarding which cohort to target during LUC in the model. (3) Model documentation is enhanced in the gmd-2017-118 paper, with dissemination aspects being strengthened. In particular, DGVMs having already implemented gross land use change have been referred to and discussed in parallel with our implementation where relevant, in response to several reviewers' comments on this aspect. (4) The reviewer raised the question of sensitivity of simulated land turnover emissions to the number of cohorts represented in the model. We think it is not the number of cohorts that matters per se, but including more than one sub-grid secondary cohorts in the model allows testing the sensitivity of emissions to the biomass (or woody mass) of forests being cleared. We tested such sensitivity for the African continent and a relationship between emissions and cleared forest biomass has been derived and included in the revised discussion section of the BG paper. (5) Following the suggestion by the 2nd review of this paper, we performed an additional S4 simulation, which includes only net land use change and wood harvest. The emissions of land turnover and wood harvest by comparing this simulation with others are discussed in the revised manuscript. This is to investigate the influence of simulations set-up on quantified land use emissions. (6) In the revised manuscript, the implication of our finding, i.e., lower emissions when taking into account age structure, is further discussed in relevance with our model implementation and the work of Arneth et al. (2017).

**Minor**

• Results of (residual) land sink (l.324-331) are confusing if not misleading. Au- thors find 89.2 PgC for 1959-2005 and compare this to the residual land sink from the global carbon budget (Le Quere et al., 2016). This addresses the ques- tion whether ORCHIDEE can simulate the land C sink as a result of changing environmental conditions, not anthropogenic LUC. This is a different question and out of scope for the present article. I suggest the paragraph l.324-331 to be dropped. Implications of higher LUC emissions simulated by models accounting for gross land use transitions as opposed to models simulating only net land use change are discussed by Arneth et al., 2017, where ORCHIDEE participated as well. This point should not repeated here.

Following the reviewer's suggestion, the lines of 324-331 are removed.

- It should be discussed that decisions with respect to priority of forest age cohorts used for conversion are unknown at the global scale.

  We have added citations of some studies over Europe (McGrath et al., 2015) on the rotational forest ages in forest management. We're not aware of global studies of such data. Following the reviewer's suggestion, this point is discussed in the revised manuscript.

- "Age classes for forest PFTs are distinguished in terms of woody biomass, while those for herbaceous PFTs are defined using soil carbon stock" (l.156): Discuss whether this definition is a problem when biomass and soil C stocks change in response to environmental conditions. I guess the simulated age distribution is therefore not an interpretable modelled quantity.

  Indeed, the cohort boundaries defined in terms of woody biomass for forests and soil C for herbaceous vegetation types are static. For forests, biomass growth curves during the spin-up simulation with a stable early 20$^{th}$ century climate and the constant preindustrial $CO_2$ concentration are used, to delineate forest cohorts corresponding to ages since the start of spin-up. We acknowledge that using such static boundaries cannot ensure exactly the same forest aboveground biomass being cleared in the transient simulation, where environmental conditions have changed in response to anthropogenic perturbations. If we assume land managers always clear forest according to their ages, then the simulated land use emissions might be underestimated. But in general, we think the uncertainties by using static cohort boundaries should be less influential than the uncertainty brought about by the fact that — globally, rotational lengths of land turnover are poorly known and we have assumed constant, biome-specific rotation lengths. These points are discussed in the revised manuscript. Because of these uncertainties, the simulated age distribution from our simulation in this study is more considered for demonstrating the model capability rather than having solid scientific significance. It is for this reason that, even though we can get such a map, it has not been presented in the paper.

- "the land turnover resulting from the upscaling of 0.5$^{°}$ to 2$^{°}$ is not included" (l.240). This can be quite substantial. When transition maps are aggregated to a lower resolution for each transition separately, then this additional land turnover should be automatically included. How come it is not?

  Land turnover activities are represented in the model using land transition matrices. These matrices are constructed during the process to reconcile LUH1 historical land-use transition data and the current-day PFT map used by ORCHIDEE. Somehow during this process the land turnover resulting from spatial upscaling is unfortunately neglected. It can be challenging to rerun all the simulations with updated land turnover matrices due to computation limitation (because using a total number of 65 cohort functional types has tripled the time needed, in comparison to a default ORCHIDEE-MICT run which is already slow due to many processes included). On the other hand, this will not change the fundamental conclusions of the current manuscript. Based on these considerations, we have re-done rebuilding the turnover matrices by including the spatial upscaling. Then we described the missing LUC areas by ignoring the gross LUC from spatial upscaling, and used this information to correct the simulated emissions.

- "Following LUH1 (Hurtt et al., 2011), we assume that no land use change occurs during the

model spin-up." (l.249). See my comment in the reviews of Yue et al. (2017), availble through https://www.geosci-model-dev-discuss.net/gmd- 2017-118/discussion, regarding model spin up: *Fig. 6 [in the GMDD paper] shows that if a constant land turnover rate is applied during the transient simulation, but not during spinup, biomass C stocks attain the "wrong" equilibrium. I.e. stocks decline after being subjected to continuous land turnover to a new steady state, reached after around 50 years (under a tropical climate). Soil C stocks likely take longer to attain a new steady state and in cold climates even more so. If simulations are evaluated from the start of the transient simulation, then land-atmosphere C fluxes related to reaching this new steady state confound results. How is this treated when, for example, doing a historical simulation starting in 1850? Shouldn't a continuous land turnover pattern be applied already during spin up in order to avoid these disequilibrium fluxes?*

We agree with the reviewer that ideally, some form of land turnover processes, or shifting cultivation should have been included during the spin-up to mimic the already existing land use activities before the start year of the simulation, which in our case is 1501. Failing to account for this may lead to a spike in generated land use emissions due to a too large initial forest biomass, as pointed by the reviewer and shown in Fig. 2 of Stocker et al. 2014 as well as in the result of Hansis et al. (2015). Surprisingly, in our results of Fig. 3, $E_{LUC\ net}$ and $E_{LUC\ turnover}$ do not show such an initial large value starting from 1501, probably due to a too small implemented LUC area. Such initial large emissions do appear in $E_{LUC\ harvest}$, which results from a distinctly larger-than-zero primary forest harvest by the forcing data, consistent with the results by Stocker et al. 2014 and Hansis et al. 2015. Overall, such an impact of not including pre-spinup land turnover in simulated ELUC is negligible in our results (Fig. 2).

On the other hand, not including net land use changes (not land turnover) prior to the start of spin-up might lead to the omission of their legacy emissions, potentially balancing the effects by omitting pre-spinup land turnover. At last, in Table 3 we made the focus on comparing simulated emissions for the period of 1850–2005, which is expected being little impacted by the absence of pre-spinup land turnover.

All these points are discussed in the revised manuscript.

• Eq. 1 (l.256): Why is this decomposition defined here but no results for separated components are shown. Is Eq. 1 really necessary?

We intend to keep Eq. 1 for a clear definition of NBP in our model. For one reason, NBP can mean different things for different models depending on processes that are included in the model (for example, wood product decomposition or crop harvest). For another reason, this has provided a clear definition for readers who are not familiar with NBP definition in the DGVMs.

• l.363-375: It's important to note that harvest data used here specifies the harvested forest area. LUH alternatively provides harvested wood mass as a forcing dataset. Results presented here are subject to this choice and to the predefined priority rules (which age cohort to harvest first). According to l.172, the same priority rules are specified for land turnover and wood harvest, that is, middle-aged forest is harvested with a priority. Is this plausible? It may at least be equally plausible to assume that the oldest patch is harvested first as it

has the highest biomass. In that case, the $S_{age}$ simulation should have higher wood harvest- related emissions and the difference to $S_{ageless}$ should be small.

We agree with the reviewer assuming the oldest forest patch being harvested first will yield higher emissions. But in practice foresters tend to maintain an optimal rotation length to maximize the profit and if we know this age for different regions of the globe, then setting the primary-target cohort with such an age in the model will make sense. This is our major motivation to include the priority decision rules in wood harvest in the model. But unfortunately, the information on the contemporary and historical forest rotation length seems to be scattered in literature and no systematic compilation of such information exists. This point will be discussed in the revised manuscript.

- l.542-543: Mention here how these compare to the un-corrected values.

  This will be done in the revised manuscript.

- l.611: What does "down-estimate" mean?

  We mean a downward shift in the revision of emissions from shifting cultivation. This is now changed to "the extent to which emissions from shifting cultivation can be revised by a downward shift".

- l. 615 (Conclusions): "This [accounting for cohorts] will lead to a lower-than- assumed so-called residual land CO2 sink on undisturbed land, which is inferred from the net balance of emissions from fossil fuel and land use change, and CO2 sinks in the atmosphere and ocean". This is a change of a change (age cohort effects on top of gross vs. net land use change effect) and the conclusion for a lower than expected residual land sink might appear confusing after Arneth et al. (2017) concluded a likely higher-than-expected residual land sink.

  As far as we know at least some models (like JSBACH and the ORCHIDEE version used there) in Arneth et al. (2017) do not account for secondary forests clearing in shifting cultivation or wood harvest. Therefore the emissions from these previously overlooked processes in DGVMs are likely overestimated there, although we agree that directional change (i.e., an upward shift in the revision of total emissions) is without any question. This point will be further clarified in the revised manuscript.

---

## Author Comment (AC2) · 26 Nov 2017

Yue and co-authors do in this paper demonstrate how inclusion of differently aged forests in the ORCHIDEE DGVM leads to reduced global carbon emissions (CE) from land use changes (LUC) during the period 1501-2005. This reduction is mainly attributed to the part of the CE which stems from shifting cultivation in the tropics (which they also included as a new feature in ORCHIDEE)). The authors systematically quantify the contribution of different processes (net LUC, shifting cultivation and wood har- vest) to the total CE from LUC (ELUC). The study is thus an important contribution to quantifying the ELUC which clearly demonstrates the importance of the inclusion of many aspects of vegetation dynamics and LUC to obtain accurate estimates of ELUC.

We thank the reviewer for the efforts to review our paper and the general comments. Please find out point-to-point response below each comment.

The paper is in general clearly written (though the authors at some places tend to repeat themselves), well structured and easy to read.

The main part of the description of the model development has been put in an accompanying paper "Representing anthropogenic gross land use change, wood harvest and forest age dynamics in a global vegetation model ORCHIDEE-MICT (r4259)", Global Model Development Discussions, 2017-118 (hereinafter GMD118), where the model functionality is demonstrated in an idealized site study and a regional study in South Africa. Since these two papers are closely related, some of my comments (including the main comment on the setup on the S-experiments) below also apply to GMD118 (unfortunately I missed the discussion deadline for GMD118).

The idea to separate the work in a development and an application part seems nice, but the separation between the two papers is not very clear: A lot of the model description is repeated in the present paper, and the analysis methods and results are very similar for the "South Africa" study and the global study which suggest to replace the results of the "South Africa" study with those of the global one in GMD118.

The other reviewer of this paper has raised similar comments regarding the likely overlap between the two manuscripts. We appreciated the suggestion and have restructured both papers to make a clearer separation between them. Please refer to our response [R3] to the other reviewer's comments on this issue.

Though the papers (present and GMD118) represent a valuable contribution to the quantification

of ELUC and its originating processes, there are a number of issues to be addressed at different level of severity:

Major:

Though qualitatively the major conclusion of the paper (effect of introducing age classes on gross transitions LUC) is obvious, unfortunately the experimental setup is not optimal for supporting this conclusion quantitatively. The authors use an "additional process approach" by starting with a model without any LUC (their S0), then adding net transitions (S1), gross transitions (also called "turnover", S2) and finally wood harvest (S3). Such an approach only delivers a best guess for the last step - i.e. the wood harvest. However the main conclusion is about the turnover and the result does thus ignore the differences in the effects of wood harvest between the different experiments, which are clearly present (e.g. their increase in ELUC_harvest from ageless to age). To provide a best guess on the effect of turnover, an additional experiment (I call it S4), including net transitions and wood harvest but ignoring turnover, would be needed. The turnover effects are then calculated from the difference between S4 and S3 instead of between S2 and S1. This could either be used to throw out S2 (the S2 setup is - to my knowledge - not used by any model, and thus is only usable to provide good estimates on the effects of wood harvest, not for model intercomparison) or to turn the general experimental structure into a "subtractive process approach", based on the "best guess" experiment (S3) and analyzing the effects of the different processes by removing them individually (turnover by comparing to S4, harvest by comparing to S2). In the first case the quality of the ELUC from wood harvest will be degraded, in the latter, some structural changes are needed to the paper.

We thank the reviewer for this thoughtful comment. We make it clear in the revised manuscript that separating the overall LUC activities into these three processes is to examine their individual contributions from a theoretical, modeling perspective, in particular given that land turnover or gross land use change has been overlooked by modeling practices (in view of Arneth et al. 2017). In reality, however, these three activities might never be clearly separated, for example, a fallow forest following agricultural abandonment in land turnover process might later be maintained for wood harvest, or vice versa. Here we followed the approach of Stocker et al. (2014) to run additive factorial simulations to quantify the effect of each process.

We followed the reviewer's suggestion to add an additional S4 simulation, which includes net transitions and wood harvest (it is named S2b simulation). Both emissions from land turnover and wood harvest are calculated from an additive and a subtractive approach. For example, for land turnover, emissions to the additive approach are quantified as the difference between S2 and S1 simulation (the original ones in the manuscript), and emissions to the subtractive approach are quantified as the difference between S3 and S2b. An overview table as below is now provided in the Supplement Material to show the emissions to different approaches (in unit of PgC for 1501–2005):

|  | No age dynamics | With age dynamics |
|---|---|---|
| ELUC_net | 123.7 | 118.0 |
| ELUC_turnover | 45.4 | 27.3 |
| ELUC_turnover_S2b | 39.9 | 25.1 |
| ELUC_harvest | 27.4 | 30.8 |
| ELUC_harvest_S2b | 32.9 | 33.0 |

As is shown in the table, different approaches have a larger impact on $E_{LUC\ turnover}$ and $E_{LUC\ harvest}$ by the $S_{ageless}$ simulations than $S_{age}$ simulations, but in general the difference between different approaches is much smaller than the emissions itself. This indicates that the impacts of LUC on carbon emissions simulated by ORCHIDEE are largely a linear system. Overall, we cannot agree with the reviewer that a subtractive approach is necessarily superior to an additive one, even for a nonlinear system. For a quasi-linear system like the case here, we think that using either approach would yield small differences. For a nonlinear system, different approaches can be used depending on the purpose of attribution being performed, sometimes a re-scaling or more complex treatment techniques might be needed (e.g., Ciais et al., 2013; Trudinger and Enting, 2005).

I don't see the added value of the "South Africa" study in GMD118 in addition to the idealized site level study (also in GMD118) and the global study presented in this paper. The description of the "South Africa" sub-study in GMD118 is very short and hardly complete (e.g. which initial vegetation distribution was used?, were the LUH1 data backcast as in the present global study?)

Following the comments by both reviewers of this paper and the reviewer's comments on gmd-2017-118, the results for the carbon fluxes of the Southern African study has been removed in the revised GMD paper, with only the results on the forest cohort dynamics (Fig. 9) being kept. This is a model feature from our developments that we would like to present. Please see our responses to the similar question raised by Benjamin Stocker on this paper [R3] and the responses in gmd-2017-118.

A reasoning and discussion of the validity and influence of the priority rules for turnover and wood harvest is absent in this paper though some discussion is included in GMD118. This needs to be added or at least referenced and could also advantageously be extended.

The primary target cohort for turnover and its age setting (15 years) mainly depends on the assumptions used in LUH1 data, as has been explained in the manuscript. We think the systematic, worldwide information on rotation lengths of shifting cultivation or wood harvest is lacking. This partly hinders our work to set a more reasonable, regionally varying target cohorts and their ages in the model. This point and its influences on derived carbon emissions are now discussed in the revised manuscript.

The authors several times mention "inconsistencies between LUH1 and ESA-CCI-LC", but these problems may as well - at least in parts - stem from the choice of priority rules and the assumptions by Hurtt et al. (2011) for creating the global LUH1 data set. At least some comments attempting to disentangleing these effects should be made.

See e.g. the discussion in Arneth et al. (2017) and references therein.

The inconsistencies between LUH1 and ESA-CCI-LC are also partly due to the fact that we used the harvested forest area, rather than the wood volume as the input information. But most of the inconsistencies are because of the spatial inconsistencies between the two land cover maps (LUH1 and ESA-CCI-LC). The choice of priority rules is at a lower hierarchical level than the distributions of, and the transitions among the land cover types. Therefore it would not cause any additional inconsistency. The inconsistencies between LUH forest area and that observed by satellites is also highlighted in Meiyappan and Jain (2012), and the inconsistencies in land use transitions among different data sets have been highlighted in Li et al. (2017).

l. 507-543: Upscaling the ELUC based on scaling the total carbon to the TRENDY intermodel mean is very speculative and does - though it seems so - not add any quantitative information - specially not since the main focus of the paper is on the effects of including (or excluding) certain processes and not on the absolute ELUC numbers. I suggest to put the entire paragraph together to (essentially, not literally): "We have low absolute ELUC, relating to a low absolute carbon stock. These two quantities seems to be linearly related (Li et al. 2017)". This let the readers do the upscaling themselves being aware that this extrapolation is only qualitatively valid. This leaves Fig. S8, Table S2 and perhaps Table S1 (the main message can also be extracted from Table 3) obsolete.

The section 4.2 addresses the errors in simulated biomass stock in the current model version and its impact on simulated $E_{LUC}$. We think this section is necessary. It is also important to show that converging values can be obtained by adjusting for such errors. We have chosen not to put such information in the main text because we share the reviewer's comments that these are not completely valid in a quantitative sense. As these are just materials in the Supplement, we tend to keep them, assuming that they can be potentially useful for readers interested to know more details on the corrections that have been made. However, we stress in the revised manuscript that such extrapolations should be taken with caution and the numbers derived are not fully quantitatively valid.

The presentation let the model development seem entirely new, though Reich et al. (2013) contains a similar introduction of gross transitions and Shevliakova et al. (2009) introduced both vegetation with different age and gross transitions. These two studies must be taken into account in the description of the model development.

Following the reviewer's suggestion, the introduction section and model description section will be revised to fully reflect these relevant peer works. We included in the revised gmd-2017-118 an overview table of DGVMs having implemented gross land use change and that table will also be referred to in the revised manuscript of BG paper.

Minor:

Are S0-S2 and the Spinup entirely without wood harvest or do they use a fixed preindustrial (1500) wood harvest? If no harvest has been used, S3 will be subject to a "carbon chock" at the beginning of the transient run stemming from starting from a wrong equilibrium state and the absolute ELUC numbers - specially from S3 - are likely overestimated (S0 contains too much

carbon).

S0 to S2 and spinup runs do not include any wood harvest or land turnover. The "carbon shock" indicated by the reviewer is visible in Fig. 3c for the few beginning years since 1501 but such an initial peak of emissions is small compared with the cumulative emissions over 1501–2005. We agree with the reviewer that omitting wood harvest in the spinup runs and in the simulations of S0 to S2 will lead to overestimation of emissions from wood harvest, but their impacts on emissions after 1850 are expected to be very small due to the fading of legacy effects with time. On the other hand, these runs do not include either the legacy emissions from net land use changes before 1501, which can lead to underestimation of emissions. These points will be discussed in the revised manuscript. Please also refer to our responses to a similar comment raised by the other reviewer of this paper (Page 5 of the responses to the comments by Benjamin Stocker).

Figure 6 needs to be introduced in paragraph 2.2 (likely with a lower number), since it actually do not show the results of the work of the authors but is rather a part of the description of the LUH1 data set. The figure is, however, absolutely necessary for the understanding of the results.

We agree with the reviewer on that Fig. 6 is not the result of our work in a very strict sense, although it is in fact an output of reconciling LUH1 data and the ORCHIDEE PFT map derived from the ESA-CCI-LC land cover map. Note that in the original LUH1 data land use transitions are not downscaled to forests or grasslands, it is after such reconciliation that historical LUC areas involving forests have been reconstructed. To put this figure as Fig. 6 allows readers to easily refer to it when going through the results of regional LUC emissions. On the other hand, introducing this figure in the section 2.2 would be a little isolated without presenting it in detail (whose details are presented rather in the section 3.3). For the reconciliation between the LUH1 data and the ORCHIDEE PFT map, all relevant outputs in section 2.2 are provided in the Supplement, which has been referred to in the section 2.2. We believe this can provide useful information if readers are interested on the specific outcomes of the historical LUC data reconstruction.

The numbers in Line 544-551 should also be introduced when introducing the LUH1 data set (paragraph 2.2). It is rather important for evaluating the results to know that substantial fractions of some of the transitions in the LUH1 data set are ignored.

Following the reviewer's suggestion, we have moved these descriptions from the section 4.2 to the revised section 2.2, which are further referred to in the revised section 4.2.

Was "apparent gross transitions" arising from the aggregation of LUH1 (which only contains gross transitions in the tropics) over multiple grid cells actively suppressed outside the tropics? If yes: Why? This seems to be an unnecessary loss of information.

Such a loss of information is not out of an intentional active suppression. It is unfortunately due to an error in upscaling the data from 0.5° to 2°. Land turnover activities are represented in the model using land transition matrices. These matrices are constructed during the process to reconcile LUH1 historical land use transition data and the current-day PFT map used by ORCHIDEE. Somehow during this process the land turnover resulting upscaling is unfortunately

neglected. It can be challenging to rerun all the simulations with updated land turnover matrices because of computation limitation (because including a total number of 65 cohort functional types has tripled the time needed, compared to a default ORCHIDEE-MICT run which is already slow due to many processes included). On the other hand, this will not change the fundamental conclusions of the current manuscript. Based on these considerations, we have re-done the process to build up the turnover matrices by including the gross land use change in spatial upscaling. Then we described the missing LUC areas by ignoring the gross LUC from spatial upscaling, and used this information to correct the simulated emissions.

The division of herbaceous vegetation into two age cohorts based on the soil carbon (SOC) is either insufficiently explained or only representative for a certain type of LUC. In line 53-54 of GMD118 the authors state: "SOC decreases when a forest is converted to cropland; SOC increases when a cropland is converted to pasture" indicating that young herbaceous vegetation can have SOC both higher and lower SOC than the previous vegetation. Furthermore it seems that the division ignores that the main part of the changes in SOC do not take place instantaneously at the time of LUC.

The key point is to separate agricultural lands (croplands and pastures) into two broad age groups assuming that they have different soil carbon stocks. In general, because changes of soil carbon stock following land use change are spatially highly diverse and depend on many factors including the land cover types before and after the transition, the model feature described here is more for informative purpose rather than having solid scientific significance. This is primarily due to the fact that soil moisture is simulated on the basis of water columns, and soil temperature over the whole grid cell in the model rather than on the cohort level, as is explained in the gmd-2017-118 paper (Sect. 2.2.3, 2nd paragraph). To fully track the soil carbon trajectory after land use change, a much larger number of cohorts for herbaceous vegetation are needed, but this is limited by the computing power when running simulation over the globe. Overall, this feature is more like a "place holder" whose function needs to be explored in the future model application.

Because of the diverse changes in SOC following land use change and the limitation in the number of cohorts, we choose a threshold of 65% of the maximum SOC of all vegetation types as the cohort boundary, hoping this can partly accommodate the large ranges in SOC in different land cover types. It turns out such setting seems not working well, as the cohort dynamics of herbaceous vegetation do not show a reasonable temporal pattern as shown in the case of forests (Fig. 9 in gmd-2017-118). Nonetheless, as the differences in land turnover emissions between the two simulations with and without sub-grid cohorts are mainly driven by sub-grid secondary forest dynamics, the influence of errors in setting herbaceous cohorts is expected to be small. All these points will be discussed in the revised manuscript.

Technical:

It should be made clear earlier in the paper that the terms "shifting cultivation" and "turnover" are used interchangeably.

This is a good point and will be done in the revised manuscript.

Please repeat the main quantitative findings of the study in the conclusions. In some cases letters

are swapped in the subscripts.

We repeat the main findings in the conclusions of the revised manuscript and subscripts are double-checked.

Figs. 4-6 and S7: Please swap the order of the sub-panels from column-wise to row- wise. This is used in Fig. 2 and is much more intuitive.

Following the reviewer's suggestion, we have revised Fig. 5–6 and Fig. S7. For Fig. 4, we keep the current layout. This figure shows in each row, the simulated $E_{LUC}$ by $S_{ageless}$, the age effect and the LUC area involved for each LUC type. Though not being supported by any serious scientific papers, we think to compare maps in a horizontal layout is more intuitive for human eyes.

Figs. 5, 6 and S7: The order of the geographical regions seems totally random. Please introduce some "around-the-globe"-ordering as in e.g. v.d.Werf et al. (2010). I am not saying, that the authors should adopt the regions from v.d.Werf - just the systematic ordering principle.

Thanks for this good suggestion. The order of regions in these figures will be re-arranged.

Figs. 3d-3f, 6, S3, S4 and S6: The unit Mkm^2 is not a valid SI unit (double prefix). Please use "Mill. km^2", "10^6 km^2", "10^12 m^2" or rescale to e.g. "MHa" (which would fit the numbers in Figs. 3 and 6 quite well).

Thanks for this good suggestion. The unit of Mkm^2 is changed to $10^6 km^2$.

Fig. 5 vs. 6: It is confusing that Fig. 5 starts in 1900 which Fig. 6 starts in 1800. The only thing mentioned in the paper before 1900 is - as far as I see - the peaks in North America. Does that need to be displayed?

Indeed, the only reason to start the horizontal axes of Fig. 6 from the year 1800 is to show the strong legacy impact on emissions in North America. Following the reviewer's suggestion, we have changed Fig. 6 to have the same horizontal axis range as Fig. 5. The pre-1900 LUC area in North America is still described but without a figure being shown.

Table 2: The main point of this table is the threshold fractions of Bmax used - the ages used for the determination are only relevant for the development stage and thus these are the numbers which should show up in brackets. Please either leave out "x Bmax" (described in the table caption) or add it everywhere - the mixture leaves the table rather confusing. The PFT-numbers are only of model internal relevance and should be removed.

We have adjusted the table to put the age information within the brackets and to put the information of fraction of $B_{max}$ in the table cells, with the meaning of "x $B_{max}$" being explained in the table caption.

In GMD118 l.477 and l.688 the LUH1 data set seems attributed to Hurtt et al. (2006) while the actual description of the data are in Hurtt et al. (2011).

In these two places the description of residence time of shifting cultivation (15 years) is cited

from Hurrt et al. (2006). Both Hurrt et al. (2006) and Hurrt et al. (2011) described these and we have cited both in the revised gmd-2018-118 paper.

The initial nomenclature is in my opinion more confusing (through unnecessary abstraction of rather simple expressions) than helpful and could be removed.

We would like to keep this nomenclature if it is allowed according to the journal policy, with the hope that it can facilitate the reading process for the readers without a specific land use change research background.

My personal opinion is that supplemental material should be kept at a minimum. For this paper this implies that the description of the backcast of the LUH1 data should rather be an appendix to the paper - or to GMD118 if the method was also applied here. Raw figure data should rather be "available upon request" than put in the supplement.

We believe the suggestion to put in an appendix the backcasting of historical land cover maps is better than putting them in the Supplement if the journal policy allows. We will check with the editorial staff of the journal on this. Several papers have put the raw data in the Supplement and we followed them, but this might not be compulsory. We will also check with the editor on this.

References:

Ciais, P., Gasser, T., Paris, J. D., Caldeira, K., Raupach, M. R., Canadell, J. G., Patwardhan, A., Friedlingstein, P., Piao, S. L. and Gitz, V.: Attributing the increase in atmospheric CO2 to emitters and absorbers, Nat. Clim. Change, 3(10), 926–930, 2013.

Li, W., MacBean, N., Ciais, P., Defourny, P., Lamarche, C., Bontemps, S., Houghton, R. A. and Peng, S.: Gross and net land cover changes based on plant functional types derived from the annual ESA CCI land cover maps, Earth Syst. Sci. Data Discuss., 1–23, doi:https://doi.org/10.5194/essd-2017-74, 2017.

Meiyappan, P. and Jain, A. K.: Three distinct global estimates of historical land-cover change and land-use conversions for over 200 years, Front. Earth Sci., 6(2), 122–139, doi:10.1007/s11707-012-0314-2, 2012.

Trudinger, C. and Enting, I.: Comparison of formalisms for attributing responsibility for climate change: Non-linearities in the Brazilian Proposal approach, Clim. Change, 68(1–2), 67–99, doi:10.1007/s10584-005-6012-2, 2005.

---

## Author Response (AR1)

The present paper presents an application of the model described in Yue et al. (2017), GMDD, for global simulations covering the period where land use change (LUC) forcing data is available (1501-2005). Simulated cumulative emissions are 118 PgC for net land use plus 27.4 PgC for effects of sub-gridscale bi-directional land turnover (shifting cultivation type agriculture) plus 30.8 PgC for effects of wood harvesting. This amounts to a total of 176 PgC. This is at the lower end of the range of available estimates.

A special focus is put on the value of distinguishing age cohorts of land patches that have been affected by land conversion at different times in the past. The paper shows that not accounting for this effect increases estimates for cumulative LUC emissions. Authors explain that this is due to the generally higher average biomass density of converted land in simulations where no age cohorts are simulated.

Since effects of land turnover (shifting cultivation) and wood harvesting have been introduced into vegetation models, it has remained unclear what effect a distinction of age cohorts would have on simulated land use change emissions. The present paper addresses this knowledge gap and presents results from two simulations - one with age cohorts distinguished ($S_{age}$) and one without ($S_{ageless}$). The reduction of the land turnover component of total emissions when comparing the two is extremes ($S_{age}$ vs. $S_{ageless}$) is 40.

This is a notable contribution to the existing literature. However, its presentation and discussion in the context of the available literature is unsatifying and some parts misleading. Moreover, the present paper has substantial overlap with Yue et al. (2017), currently under review in GMDD. These aspects should carefully be adressed in the next revision round. Below I'm listing these two major points and a few (a bit more) minor ones.

[R1] We thank the reviewer for the general positive comments and the efforts to review both papers. Please see our point-to-point responses as below. Major revised texts are tracked in the updated manuscript.

**Major**

- The point that the presentation and discussion of results in the context of the available literature is unsatifying echoes critique raised in the reviews of Yue et al. (2017), availble through https://www.geosci-model-dev-discuss.net/gmd- 2017-118/discussion, in particular the comment by J. Nabel. The same applies to the present paper. Particular attention should be paid to discuss results in the face of findings by Arneth et al. (2017) and to accurately describe which of the previously published models account for age cohorts within non-agricultural land and how many cohorts are distinguished. An overview table would help. Authors describe the $S_{age}$ simulation as reflecting the "traditional approach" (l.181), implying that the age cohort distinction is itself a novelty. However, it is not. Already Shevliakova et al. (2009) distinguished multiple cohorts. Stocker et al. (2014) distinguished two cohorts (primary and secondary land). Only the model described in Reick et al. (2013) and applied by Wilkenskjeld et al. (2014) makes no distinction between age cohorts. The LPJ-GUESS model (Smith et al., 2014) explicitly tracks C pools of land patches (cohorts) subjected to stochastic disturbance. $S_{ageless}$ thus reflects an arguably extreme case and is not reflective of any "traditional approach". Having said that, an improved introduction and discussion will address this concern.

[R2] We thank the reviewer for pointing to these studies and this greatly helps to expand the context and discussions of our work. The introduction and discussion sections in the updated manuscript have been revised to take account into these studies. In response to the reviewer's request, an overview table on current implementations of gross land use change in DGVMs is provided in the revised GMD manuscript as we think it's more appropriate there (appended at the end of this document). This overview table will be cited in the revised BG manuscript. We also invite the editor and interested readers to check the interactive discussions of the gmd-2017-118 paper (https://www.geosci-model-dev-discuss.net/gmd-2017-118/) as the reviewer's comments are highly related in these two papers, so are our responses.

- My second major concern concerns the overlap with Yue et al. 2017, where the model applied here is described more extensively. Although authors only refer to their "idealized site-scale simulations" presented in Yue et al. (2017), it should be noted that also regional scale simulations, covering southern Africa, are presented therein and the main conclusion of that paper is identical to the main conclusion of the present paper - namely that accounting for age cohorts reduces the land turnover effect contribution to total LUC emissions. I raised this issue also as a reviewer for the GMDD paper and wrote:

*The present paper [GMDD] was submitted on 14 May 2017. On 26 July 2017, Yue, Ciais and Li submitted a paper to Biogeosciences Discussions (https://www.biogeosciences-discuss.net/bg-2017-329/), where the same model is applied to investigate essentially the same questions, but this time at the global scale. The regional focus of the present paper on southern Africa may appear arbitrary at first, but makes sense. Apparently, authors preferred to devote a full paper to model description and evaluation and a second full paper to a global ap- plication. In my view, this is a viable way to go and the large work that went into developing this model warrants two separate papers. However, I find the delineation of their respective scope a bit unsatisfying. Readers will likely be left asking themselves why authors didn't present results from global simulations in the present (GMDD) paper - a relatively small additional step in terms of additional work. Simultaneously, readers of the BGD paper might be left wondering what the additional insight of that paper is after already the GMDD paper concluded that accounting for separate age cohorts reduces the effect of gross versus net LUC emissions.*

The same issue applies vice-versa, i.e. to the present (BGD) paper. I further suggested to reinforce the value of the GMDD paper in terms of its model documentation and dissemination aspects. The present paper could for example gain in its value if the age-cohort effect is investigated not only for the two extremes (1 and 6 cohorts) but for additional numbers of cohorts, to establish a functional relationship between the number of cohorts and emissions. This would address also my previous point and would allow for a better comparison with models that distinguish between primary and secondary land (2 cohorts). Of course, this is just a suggestion, but I do encourage that the authors find a solution to finding a better delineation between their parallel submissions currently under review here and in GMDD.

[R3] In view of the reviewer's comments here, and the comments on our parallel gmd-2017-118 paper, we revised both papers to make a clearer delineation in their scopes: (1) Scopes are clearly defined in the introduction of each paper. The gmd-2017-118 paper focuses on model documentation and examination / illustration of model behaviour; the current paper focuses on model application on a global scale and comparisons of simulated LUC emissions with other studies. (2) The figure on the carbon fluxes for Southern Africa in gmd-2017-118 has been removed. Only the Fig. 9 is kept there to illustrate the cohort dynamics with land use change in view of the hierarchical decision rules regarding which cohort to target during LUC in the model. (3) Model documentation is enhanced in the gmd-2017-118 paper. In particular, DGVMs having already implemented gross land use change have been referred to and discussed in parallel with our implementation where relevant, in response to several reviewers' comments on this aspect. (4) The reviewer raised the question of sensitivity of simulated land turnover emissions to the number of cohorts represented in the model. We agree that the number of cohorts matters, but more precisely and directly, including more than one sub-grid secondary cohorts in the model allows testing the sensitivity of emissions to the biomass (or woody mass) of forests being cleared. We conducted a sensitivity test in the African continent as an example, and a relationship between emissions and cleared forest biomass has been derived and included in the revised discussion section of the BG paper. (5) Following the suggestion by the 2nd review of this paper, we performed an additional S2b simulation, which includes only net land use change and wood harvest. The emissions of land turnover and wood harvest by comparing this simulation with others are discussed in the revised manuscript. This is to investigate the influence of simulations set-up on quantified land use emissions. (6) In the revised manuscript, the implication of our finding, i.e., lower emissions when taking into account age structure, is further discussed in relevance with our model implementation, the shifting cultivation rotation lengths and the associated uncertainties.

**Minor**

- Results of (residual) land sink (l.324-331) are confusing if not misleading. Au- thors find 89.2 PgC for 1959-2005 and compare this to the residual land sink from the global carbon budget (Le Quere et al., 2016). This addresses the ques- tion whether ORCHIDEE can simulate the land C sink as a result of changing environmental conditions, not anthropogenic LUC. This is a different question and out of scope for the present article. I suggest the paragraph l.324-331 to be dropped. Implications of higher LUC emissions simulated by models accounting for gross land use transitions as opposed to models simulating only net land use change are discussed by Arneth et al., 2017, where ORCHIDEE participated as well. This point should not repeated here.

  Following the reviewer's suggestion, the lines of 324-331 are removed.

- It should be discussed that decisions with respect to priority of forest age cohorts used for conversion are unknown at the global scale.

  *Following the review's suggestion, we added in the revised introduction: "In view of the fact that worldwide, systematic information on historical and present rotation lengths of shifting cultivation and wood harvest is missing, some reconstructions of land use change, such as the land-use harmonization version 1 (LUH1) data assumed a fixed rotation length of 15 years for shifting agriculture in the tropics, and this assumption has been used in some modeling studies (Bayer et al., 2017).*" The discussion on the uncertainty of historical wood harvest data, and our estimated $E_{LUC}$ has also been improved. Please refer to the revised Sect. 2.1 and Sect. 4.1.

- "Age classes for forest PFTs are distinguished in terms of woody biomass, while those for herbaceous PFTs are defined using soil carbon stock" (l.156): Discuss whether this definition is a problem when biomass and soil C stocks change in response to environmental conditions. I guess the simulated age distribution is therefore not an interpretable modelled quantity.

  *These boundaries are indeed static. To explain the implications of such a choice, we added in Sect. 2.3.2 the following texts:*

  *"We acknowledge that using such static woody biomass boundaries cannot ensure the exactly a forest of a given age to be cleared in the transient simulation, because changes in environmental conditions (e.g., atmospheric $CO_2$ concentrations, climate) may alter the woody biomass-age curves established from the spin-up results, i.e. the boundary biomass limit is reached at a younger age in case productivity increases from environmental condition changes. If we assume that land managers always clear forest according to their ages, then our simulated land use emissions might be underestimated, provided a higher biomass for a given age in transient simulations than for the spin-up state. But in general the uncertainties of using static biomass boundaries for forest cohorts should be less influential than the uncertainty brought about by the fact that — globally, rotational lengths of land turnover are poorly known and we have assumed a constant 15-year rotation length for shifting agriculture in tropical regions. For wood harvest, we also assumed three different simple fixed rotation lengths for boreal, temperate and tropical regions, respectively (Table 2)".*

  *Because of these uncertainties, the simulated age distribution from our simulation in this study is more considered for demonstrating the model capability rather than having solid scientific significance. It is for this reason that, even though we can produce a map of the age of secondary land, it has not been presented in the paper.*

- "the land turnover resulting from the upscaling of $0.5^{\circ}$ to $2^{\circ}$ is not included" (l.240). This can be quite substantial. When transition maps are aggregated to a lower resolution for each transition separately, then this additional land turnover should be automatically included. How come it is not?

  *Land turnover activities are represented in the model using land transition matrices. These*

matrices are constructed during the process to reconcile LUH1 historical land-use transition data and the current-day PFT map used by ORCHIDEE. Somehow during this process the land turnover resulting from spatial upscaling is unfortunately neglected. It is challenging to rerun all the simulations with updated land turnover matrices due to computation limitation (because using a total number of 65 cohort functional types has tripled the time needed, in comparison to a default ORCHIDEE-MICT run which is already long due to many processes being included on a 30 min time step). On the other hand, this will not change the fundamental conclusions of the current manuscript. Based on these considerations, we have re-rebuilt the turnover matrices by including the spatial upscaling. Then we described the missing LUC areas by ignoring the gross LUC from spatial upscaling. We did not provide a further correction of $E_{\text{LUC turnover}}$ by accounting for this because it does not add further credibility on our estimation. This issue is briefly described in the revised method Sect. 2.2, with the following sentences being added: "*The missing land turnover areas represent 17% of the turnover between natural lands and cropland that are included in our study, and 14% of the turnovers between natural lands and pasture. The influence of this spatial aggregation error on derived emissions will be discussed in the discussion section.*"

- "Following LUH1 (Hurtt et al., 2011), we assume that no land use change occurs during the model spin-up." (l.249). See my comment in the reviews of Yue et al. (2017), availble through https://www.geosci-model-dev-discuss.net/gmd- 2017-118/discussion, regarding model spin up: *Fig. 6 [in the GMDD paper] shows that if a constant land turnover rate is applied during the transient simulation, but not during spinup, biomass C stocks attain the "wrong" equilibrium. I.e. stocks decline after being subjected to continuous land turnover to a new steady state, reached after around 50 years (under a tropical climate). Soil C stocks likely take longer to attain a new steady state and in cold climates even more so. If simulations are evaluated from the start of the transient simulation, then land-atmosphere C fluxes related to reaching this new steady state confound results. How is this treated when, for example, doing a historical simulation starting in 1850? Shouldn't a continuous land turnover pattern be applied already during spin up in order to avoid these disequilibrium fluxes?*

    We agree with the reviewer that ideally, some form of land turnover processes should have been included during the spin-up to mimic the already existing land use activities before the start year of the simulation. Failing to account for this may lead to a spike in generated land use emissions due to a too large initial forest biomass, as pointed by the reviewer. Surprisingly, in our results of Fig. 3, $E_{\text{LUC net}}$ and $E_{\text{LUC turnover}}$ do not show such an initial large value starting from 1501, probably due to a too small LUC area. The initial large emissions due to spin-up without harvest do appear in $E_{\text{LUC harvest}}$, which results from a distinctly larger-than-zero primary forest harvest in the forcing data, consistent with the results by Stocker et al. (2014) and Hansis et al. (2015). Overall, such an impact of not including pre-spinup land turnover in simulated ELUC is negligible in our results (Fig. 2). In Table 3 we made the focus on comparing simulated emissions for the period of 1850–2005, which is expected to be little impacted by the absence of pre-spinup land turnover.

    In the revised manuscript, we added following sentences in Sect. 2.3.1: "*Following LUH1 (Hurtt et al., 2011), we assume that no land use change occurs during the model spin-up.*

*This might lead to overestimation of $E_{LUC}$ for the beginning years of the transient simulation due to high carbon stocks that are free from LUC activities before 1501. But on the other hand, legacy emissions from LUC activities before 1501 are also omitted. In general, because the magnitude of annual LUC activities for 1501–1520 is very small (data shown in Fig. 2), we assume the bias of LUC emissions induced by not including LUC in the spin-up is small. Besides, simulated $E_{LUC}$ is less influenced by this factor after ca. 1700, which dominates the total LUC emissions since 1501."*

We added further the following sentences in the discussion Sect. 4.1: *"We do not account for any LUC activities in the spin-up run and pristine ecosystems are assumed at the beginning of the transient run in 1501. This set-up might cause a spike in emissions during the beginning years in the transient simulation because ecosystem biomass stocks are high, due to a lack of historical disturbance. Such a spike was evident in results by Stocker et al. (2014, blue and green lines in their Fig. 2) when land turnover is not accounted for during the spin-up in some of their simulations. The similar model behaviour also presents in the results by Hansis et al. (2015, dark and light blue lines in their Fig. 4) using a bookkeeping model. In our study, a similar initial spike in $E_{LUC}$ shortly after 1501 is almost invisible for the net land use change and land turnover (Fig. 2a–b), probably owing to very small magnitudes of LUC area within the few years after 1501 (Fig. 2d–e). However, there is a clear peak in $E_{LUC\ turnover}$ around 1520s (Fig. 2c), a likely impact of ignoring spin-up LUC process, given that a significantly larger-than-zero harvest area is prescribed for this period (Fig. 2f). In general, the impacts of not including LUC in the spin-up process seem to be small in our results. This issue impacts much less the comparisons focusing on emissions starting from 1850 in Table 3."*

- Eq. 1 (l.256): Why is this decomposition defined here but no results for separated components are shown. Is Eq. 1 really necessary?

  We intend to keep Eq. 1 for a clear definition of NBP in our model. For one reason, NBP can have different component fluxes for different models depending on processes that are included (for example, wood product decomposition or crop harvest). For another reason, this has provided a clear definition for readers who are not familiar with NBP definition in DGVMs.

- l.363-375: It's important to note that harvest data used here specifies the harvested forest area. LUH alternatively provides harvested wood mass as a forcing dataset. Results presented here are subject to this choice and to the predefined priority rules (which age cohort to harvest first). According to l.172, the same priority rules are specified for land turnover and wood harvest, that is, middle-aged forest is harvested with a priority. Is this plausible? It may at least be equally plausible to assume that the oldest patch is harvested first as it has the highest biomass. In that case, the $S_{age}$ simulation should have higher wood harvest- related emissions and the difference to $S_{ageless}$ should be small.

  We agree with the reviewer on that assuming the oldest forest patch being harvested in priority will yield higher emissions. But in practice foresters tend to maintain an optimal rotation length to maximize profit and if we know this age for different regions of the globe, then setting the primary target cohort with such an age depending on economic demand for wood in the model will make sense. We followed the suggestion of the other review to perform another S2b simulation where wood harvest, rather than land turnover, is first added on top of net land use change. $E_{LUC\ harvest}$ quantified by differing S1 and S2b simulations are mainly driven by harvesting primary forests, and the derived emissions are similar between $S_{age}$ and $S_{ageless}$ simulations. Relevant results are included in Sect. 3.1. We added the following texts in the Sect. 4.1: "*From the S2b simulations where wood harvest, instead of land turnover, is added on top of net land use change, $E_{LUC\ harvest}$ derived from $S_{age}$ and $S_{ageless}$ are very similar because in both simulations, forests with biomass close to the one of primary forests are harvested and their carbon stocks are similar between $S_{age}$ and $S_{ageless}$. Finally, it should also be noted that reconstructions of forest wood harvest are highly uncertain. For example, LUH1 data provides a total wood harvest amount of 102 Pg C for 1850–2005 over forest and non-forest areas, whereas Houghton and Nassikas (2017) estimated as 130 Pg C. Our estimates of $E_{LUC\ harvest}$ using different approaches is 22.5–27.8 Pg for 1850–2005, close to the estimated 25.3 Pg C for 1850–2015 by Houghton and Nassikas (2017).*"

- l.542-543: Mention here how these compare to the un-corrected values.

  We changed the original sentence to the following one: "*The biomass-corrected global cumulative $E_{LUC}$ for 1850–2005 are 174–207 Pg C for the $S_{ageless}$ simulation, and 161–194 Pg C for the $S_{age}$ simulation (Table S1), larger by 10–30% than the original values.*"

- l.611: What does "down-estimate" mean?

  We mean a downward shift in the revision of emissions from shifting cultivation. This is now replaced by improved texts in the conclusion section to reflect the revisions done in the review process.

- l. 615 (Conclusions): "This [accounting for cohorts] will lead to a lower-than- assumed so-called residual land CO2 sink on undisturbed land, which is inferred from the net balance of emissions from fossil fuel and land use change, and CO2 sinks in the atmosphere and ocean". This is a change of a change (age cohort effects on top of gross vs. net land use change effect) and the conclusion for a lower than expected residual land sink might appear confusing after Arneth et al. (2017) concluded a likely higher-than-expected residual land sink.

  This further implication of our work on the inferred residual land sink is considered a little over-extended and has been removed from the conclusion section considering the new analyses done in the review process. Instead, we summarized how our results are relevant with the model assumptions and how rotation lengths can impact the estimated $E_{LUC}$ and the related uncertainties. Please refer to the revised Conclusion section for more details.


Table: An over view of DGVMs having implemented gross land use change (shifting cultivation) and forest wood harvest.

| Model name | Reference | Shifting cultivation | Wood harvest | Number of vegetation types | Number of secondary land tiles | Secondary vegetation types |
|---|---|---|---|---|---|---|
| LM3V | Shevliakova et al., 2009 | Yes | Yes | Crop, pasture, primary and secondary vegetation | Up to in total 12 tiles | Dynamic secondary vegetation type according to the total biomass and prevailing climate |
| ISAM | Jain et al., 2013; Song et al., 2016 | No | Yes | 20 PFTs: 10 forests, 2 pastures, 2 grasses, 2 savanna, 1 shrubland, 1 tundra, 2 crops | 1 tile for each secondary forest type | Tropical evergreen and deciduous forests, temperate evergreen and deciduous forests, and boreal forest |
| VISIT | Kato et al., 2013 | Yes | Yes | 14 PFTs: 8 forests/woodlands, 1 savanna, 1 grassland, 2 shrublands, 1 tundra and 1 cropland | 1 tile for each secondary PFT | 13 natural PFTs |
| JSBACH | Reick et al., 2013 | Yes | Yes | 12 PFTs: 4 forests, 2 shrubs, 2 grasslands, 2 pastures, and 2 croplands | No separate secondary lands | |
| LPX-Bern 1.0 | Stocker et al., 2014 | Yes | Yes | 10 PFTs: 8 woody, 2 herbaceous | 1 tile for each PFT | 10 PFTs |
| LPJ-GUESS | Bayer et al., 2017 | Yes | Yes[*] | 9 natural woody PFTs, 2 natural grass PFTs; 3 cropland cohort functional types, 2 pasture PFTs | 1 tile per newly created secondary land | Dynamic vegetation type according to prevailing climate and PFT competition |
| ORCHIDEE-MICT v8.4.2 | This study | Yes | Yes | 14 PFTs: 8 forests, 2 grasslands, 2 pastures and 2 croplands | Number of tiles parameterizable for each PFT | 14 PFTs |

Yue and co-authors do in this paper demonstrate how inclusion of differently aged forests in the ORCHIDEE DGVM leads to reduced global carbon emissions (CE) from land use changes (LUC) during the period 1501-2005. This reduction is mainly attributed to the part of the CE which stems from shifting cultivation in the tropics (which they also included as a new feature in ORCHIDEE)). The authors systematically quantify the contribution of different processes (net LUC, shifting cultivation and wood har- vest) to the total CE from LUC (ELUC). The study is thus an important contribution to quantifying the ELUC which clearly demonstrates the importance of the inclusion of many aspects of vegetation dynamics and LUC to obtain accurate estimates of ELUC.

We thank the reviewer for the efforts to review our paper and the general positive comments. Please find out point-to-point response below each comment. All the major revised texts are tracked in the updated manuscript.

The paper is in general clearly written (though the authors at some places tend to repeat themselves), well structured and easy to read.

The main part of the description of the model development has been put in an accompanying paper "Representing anthropogenic gross land use change, wood harvest and forest age dynamics in a global vegetation model ORCHIDEE-MICT (r4259)", Global Model Development Discussions, 2017-118 (hereinafter GMD118), where the model functionality is demonstrated in an idealized site study and a regional study in South Africa. Since these two papers are closely related, some of my comments (including the main comment on the setup on the S-experiments) below also apply to GMD118 (unfortunately I missed the discussion deadline for GMD118).

The idea to separate the work in a development and an application part seems nice, but the separation between the two papers is not very clear: A lot of the model description is repeated in the present paper, and the analysis methods and results are very similar for the "South Africa" study and the global study which suggest to replace the results of the "South Africa" study with those of the global one in GMD118.

The other reviewer of this paper (Benjamin Stocker) has raised similar comments regarding the likely overlap between the two manuscripts. We appreciated the suggestion and have restructured both papers to make a clearer separation between them. Please refer to our response to Benjamin Stocker's comments on this issue (the response numbered as "R3", on Page 3 in our response to Benjamin Stocker's comments).

Though the papers (present and GMD118) represent a valuable contribution to the quantification of ELUC and its originating processes, there are a number of issues to be addressed at different level of severity:

Major:

Though qualitatively the major conclusion of the paper (effect of introducing age classes on gross transitions LUC) is obvious, unfortunately the experimental setup is not optimal for supporting this conclusion quantitatively. The authors use an "additional process approach" by starting with a model without any LUC (their S0), then adding net transitions (S1), gross transitions (also called "turnover", S2) and finally wood harvest (S3). Such an approach only delivers a best guess for the last step - i.e. the wood harvest. However the main conclusion is about the turnover and the result does thus ignore the differences in the effects of wood harvest between the different experiments, which are clearly present (e.g. their increase in ELUC_harvest from ageless to age). To provide a best guess on the effect of turnover, an additional experiment (I call it S4), including net transitions and wood harvest but ignoring turnover, would be needed. The turnover effects are then calculated from the difference between S4 and S3 instead of between S2 and S1. This could either be used to throw out S2 (the S2 setup is - to my knowledge - not used by any model, and thus is only usable to provide good estimates on the effects of wood harvest, not for model intercomparison) or to turn the general experimental structure into a "subtractive process approach", based on the "best guess" experiment (S3) and analyzing the effects of the different processes by removing them individually (turnover by comparing to S4, harvest by comparing to S2). In the first case the quality of the ELUC from wood harvest will be degraded, in the latter, some structural changes are needed to the paper.

We thank the reviewer for this thoughtful comment. We make it clear in the revised manuscript that separating the overall LUC activities into these three processes is to examine their individual contributions from a theoretical, modeling perspective, in particular given that land turnover or gross land use change has been overlooked by past modeling practices (in view of Arneth et al. 2017). In reality, however, these three activities might never be clearly separated, for example, a fallow forest following agricultural abandonment in land turnover process might later be maintained for wood harvest, or vice versa. Here we followed the approach of Stocker et al. (2014) to run additive factorial simulations to quantify the effect of each process.

We nevertheless followed the reviewer's suggestion to add an additional 'S4' simulation, which includes net transitions and wood harvest (it is named S2b simulation). Both emissions from land turnover and wood harvest are calculated from an additive and a subtractive approach. The original Table 1 was updated to include the S2b simulations. We replaced the original Fig. 1 by a table (shown below), which gives $E_{LUC}$ from different processes quantified by various approaches.

Table 3 LUC emissions for 1501–2005 (Pg C) from different processes quantified by different approaches (see Table 1 for detailed calculations of various ELUC).

| | No age dynamics | With age dynamics | Emission change in Sage relative to Sageless (%) |
|---|---|---|---|
| $E_{LUC\ net}$ | 123.7 | 118.0 | -4.6% |

| | | | |
|---|---|---|---|
| $E_{\text{LUC turnover}}$ | 45.4 | 27.3 | -40% |
| $E_{\text{LUC turnover S2b}}$ | 39.9 | 25.1 | -37% |
| $E_{\text{LUC harvest}}$ | 27.4 | 30.8 | 12% |
| $E_{\text{LUC harvest S2b}}$ | 32.9 | 33.0 | 0.0% |
| $E_{\text{LUC total}}$ | 196.5 | 176.1 | 10% |

As is shown in the table, different approaches have a larger impact on $E_{\text{LUC turnover}}$ and $E_{\text{LUC harvest}}$ in the $S_{\text{ageless}}$ simulations compared to $S_{\text{age}}$ simulations, but in general the difference between different approaches is much smaller than the emissions itself (~10% of the mean value between the two simulations). This indicates that the impacts of LUC processes on carbon emissions simulated by ORCHIDEE are largely linear (additive). Overall, we cannot agree with the reviewer that a subtractive approach is necessarily superior to an additive one, even for a nonlinear system. For a quasi-linear system like the case here, we think that using either approach would yield small differences. For a nonlinear system, different approaches can be used depending on the purpose of attribution being performed, sometimes a re-scaling or more complex treatment techniques might be needed (e.g., Ciais et al., 2013; Trudinger and Enting, 2005).

Following these new simulations and analyses, the relevant sections in methods (Sect. 2.3.1), results (Sect. 3.1) and discussions (Sect. 4.1) are revised accordingly in the updated manuscript.

I don't see the added value of the "South Africa" study in GMD118 in addition to the idealized site level study (also in GMD118) and the global study presented in this paper. The description of the "South Africa" sub-study in GMD118 is very short and hardly complete (e.g. which initial vegetation distribution was used?, were the LUH1 data backcast as in the present global study?)

Following the comments by both reviewers of this paper and the reviewer's comments on gmd-2017-118, the results for the carbon fluxes of the Southern African study has been removed in the revised GMD paper, with only the results on the forest cohort dynamics (Fig. 9) being kept. This is a model feature from our developments that we would like to present. Please see our responses to the similar question raised by Benjamin Stocker on this paper (the response numbered as "R3" on Page 3 in our response to Benjamin Stocker's comments) and the responses in gmd-2017-118 (https://www.geosci-model-dev-discuss.net/gmd-2017-118/).

A reasoning and discussion of the validity and influence of the priority rules for turnover and wood harvest is absent in this paper though some discussion is included in GMD118. This needs to be added or at least referenced and could also advantageously be extended.

The primary target cohort for turnover and its age setting (15 years) mainly depends on the assumptions used in LUH1 data, as has been explained in the manuscript. We think that the systematic, worldwide information on rotation lengths of shifting cultivation or wood harvest is lacking. This partly hinders our work to set a more reasonable, regionally varying target cohorts and their ages in the model. This point is discussed in an enhanced manner in the revised GMD paper. Further, we added in the revised introduction section in this paper: "*In view of the fact that worldwide, systematic information on historical and present rotation lengths of shifting cultivation and wood harvest is missing, some reconstructions of land use change, such as the*

*land-use harmonization version 1 (LUH1) data assumed a fixed rotation length of 15 years for shifting agriculture in the tropics, and this assumption has been used in some modeling studies (Bayer et al., 2017).*" To investigate the impacts of rotation lengths and the associated primary target cohorts on the estimated $E_{LUC}$, we have run a set of simulations over Africa. A new second paragraph in Sect. 4.1 is added to address this issue.

The authors several times mention "inconsistencies between LUH1 and ESA-CCI-LC", but these problems may as well - at least in parts - stem from the choice of priority rules and the assumptions by Hurtt et al. (2011) for creating the global LUH1 data set. At least some comments attempting to disentangleing these effects should be made. See e.g. the discussion in Arneth et al. (2017) and references therein.

The inconsistencies between LUH1 and ESA-CCI-LC are also partly due to the fact that we used the harvested forest area, rather than the wood volume as the input information. But most of the inconsistencies are because of the spatial inconsistencies between the two land cover maps (LUH1 and ESA-CCI-LC). The choice of priority rules is at a lower hierarchical level than the distributions of, and the transitions among the land cover types. Therefore it would not cause any additional inconsistency. The inconsistencies between LUH forest area and that observed by satellites is also highlighted in Meiyappan and Jain (2012), and the inconsistencies in land use transitions among different data sets have been highlighted in Li et al. (2017). To briefly discuss this point, we added in Sect. 2.2, "*Such inconsistencies among different data sets are a rather common challenge for their application in DGVMs, which have been reported by, for example, in Li et al. (2017a), Meiyappan and Jain (2012) and Peng et al. (2017).*"

l. 507-543: Upscaling the ELUC based on scaling the total carbon to the TRENDY intermodel mean is very speculative and does - though it seems so - not add any quantitative information - specially not since the main focus of the paper is on the effects of including (or excluding) certain processes and not on the absolute ELUC numbers. I suggest to put the entire paragraph together to (essentially, not literally): "We have low absolute ELUC, relating to a low absolute carbon stock. These two quantities seems to be linearly related (Li et al. 2017)". This let the readers do the upscaling themselves being aware that this extrapolation is only qualitatively valid. This leaves Fig. S8, Table S2 and perhaps Table S1 (the main message can also be extracted from Table 3) obsolete.

The section 4.2 addresses the errors in simulated biomass stock in the current model version and its impact on simulated $E_{LUC}$. We think this section is necessary. It is also important to show that converging values can be obtained by adjusting for such errors. We have chosen not to put such information in the main text because we share the reviewer's comments that these are not completely valid in a quantitative sense. As these are materials in the Supplement and only optional for interested readers, we tend to keep them, assuming that they can provide more details on the corrections that have been made. However, we stress in the revised manuscript that such extrapolations should be taken with caution and the numbers derived are not fully quantitatively valid. We added at the end of the 3rd paragraph of Sect. 4.2: "*However, these corrected values should be taken with caution and they're not fully quantitatively valid.*"

The presentation let the model development seem entirely new, though Reich et al. (2013) contains a similar introduction of gross transitions and Shevliakova et al. (2009) introduced both vegetation with different age and gross transitions. These two studies must be taken into account in the description of the model development.

Following the reviewer's suggestion, the introduction section is substantially revised to account for the previous relevant studies. We included in the revised gmd-2017-118 an overview table of DGVMs having implemented gross land use change and that table is also referred to in the revised manuscript of BG paper.

Minor:

Are S0-S2 and the Spinup entirely without wood harvest or do they use a fixed preindustrial (1500) wood harvest? If no harvest has been used, S3 will be subject to a "carbon chock" at the beginning of the transient run stemming from starting from a wrong equilibrium state and the absolute ELUC numbers - specially from S3 - are likely overestimated (S0 contains too much carbon).

S0 to S2 and spinup runs do not include any wood harvest or land turnover. The "carbon shock" indicated by the reviewer is visible in Fig. 3c for the few beginning years since 1501 but such an initial peak of emissions is small compared with the cumulative emissions over 1501–2005. We agree with the reviewer that omitting wood harvest in the spinup runs and in the simulations of S0 to S2 will lead to overestimation of emissions from wood harvest, but their impacts on emissions after 1850 are expected to be very small due to the fading of legacy effects with time. On the other hand, these runs do not include either the legacy emissions from net land use changes before 1501, which would lead to underestimation of emissions. These points are discussed in the revised manuscript, in the 1$^{st}$ paragraph of Sect. 2.3.1, and the last paragraph of Sect. 4.1. Please also refer to our responses to a similar comment raised by the other reviewer of this paper (Page 5 of the responses to the comments by Benjamin Stocker).

Figure 6 needs to be introduced in paragraph 2.2 (likely with a lower number), since it actually do not show the results of the work of the authors but is rather a part of the description of the LUH1 data set. The figure is, however, absolutely necessary for the understanding of the results.

We agree with the reviewer on that Fig. 6 is not the result of our work in a very strict sense, although it is in fact an output of reconciling LUH1 data and the ORCHIDEE PFT map derived from the ESA-CCI-LC land cover map. Note that in the original LUH1 data land use transitions are not downscaled to forests or grasslands, it is after such reconciliation that historical LUC areas involving forests have been reconstructed. To put this figure as Fig. 6 allows readers to easily refer to it when going through the results of regional LUC emissions presented in Fig. 5. On the other hand, introducing this figure in the section 2.2 would be a little isolated if it is not presented in detail (whose details are presented rather in the section 3.3). For the reconciliation between the LUH1 data and the ORCHIDEE PFT map, all relevant outputs in section 2.2 are provided in the Supplement, which has been referred to in the section 2.2. We believe this can already provide sufficiently useful information if readers are interested on the specific outcomes of the historical LUC data reconstruction.

The numbers in Line 544-551 should also be introduced when introducing the LUH1 data set (paragraph 2.2). It is rather important for evaluating the results to know that substantial fractions of some of the transitions in the LUH1 data set are ignored.

Following the reviewer's suggestion, we have moved these descriptions from the section 4.2 to the revised section 2.2, which are further referred to in the revised section 4.2. We have further discussed the consequences on $E_{LUC}$ by omitted LUC transitions. Please refer to the last paragraph of the revised Sect. 4.2.

Was "apparent gross transitions" arising from the aggregation of LUH1 (which only contains gross transitions in the tropics) over multiple grid cells actively suppressed outside the tropics? If yes: Why? This seems to be an unnecessary loss of information.

Such a loss of information is not out of an intentional active suppression. It is unfortunately due to an aggregation error in upscaling the data from 0.5° to 2°. Land turnover activities are represented in the model using land transition matrices. These matrices are constructed during the process to reconcile LUH1 historical land use transition data and the current-day PFT map used by ORCHIDEE. Somehow during this process the land turnover resulting upscaling is unfortunately neglected. It can be challenging to rerun all the simulations with updated land turnover matrices because of computation limitation (because including a total number of 65 cohort functional types has tripled the time needed, compared to a default ORCHIDEE-MICT run which is already slow due to many processes being included). On the other hand, this will not change the fundamental conclusions of the current manuscript. Based on these considerations, we have re-done the process to build up the turnover matrices by including the gross land use change in spatial upscaling. Then we described the missing LUC areas by ignoring the gross LUC from spatial upscaling. We did not provide a further correction of $E_{LUC\ turnover}$ by accounting for this because it does not add more credibility on our estimation. This issue is briefly described in the revised method Sect. 2.2, with the following sentences being added: "*The missing land turnover areas represent 17% of the turnover between natural lands and cropland that are included in our study, and 14% of the turnovers between natural lands and pasture. The influence of this spatial aggregation error on derived emissions will be discussed in the discussion section.*"

The division of herbaceous vegetation into two age cohorts based on the soil carbon (SOC) is either insufficiently explained or only representative for a certain type of LUC. In line 53-54 of GMD118 the authors state: "SOC decreases when a forest is converted to cropland; SOC increases when a cropland is converted to pasture" indicating that young herbaceous vegetation can have SOC both higher and lower SOC than the previous vegetation. Furthermore it seems that the division ignores that the main part of the changes in SOC do not take place instantaneously at the time of LUC.

The key point is to separate agricultural lands (croplands and pastures) into two broad age groups assuming that they have different soil carbon stocks. In general, because changes of soil carbon stock following land use change are spatially highly diverse and depend on many factors including the land cover types before and after the transition, the model feature described here is more for informative purpose rather than having solid scientific significance. This is primarily due to the fact that soil moisture is simulated on the basis of water columns, and soil temperature over the whole grid cell in the model rather than on the cohort level, as is explained in the gmd-2017-118 paper (Sect. 2.2.3, 2nd paragraph). To fully track the soil carbon trajectory after land use change, a much larger number of cohorts for herbaceous vegetation are needed, but this is limited by the computing power when running simulation over the globe. Overall, this feature is more like a "place holder" whose function needs to be explored in the future model application. These points are explained in the revised GMD manuscript (Sect. 2.1.3, the 4[th] paragraph, and Sect. 2.2.3, the last paragraph).

In the revised manuscript of the current paper, we added following sentences in the 4[th] paragraph of Sect. 2.1 to clarify these points: "*For herbaceous PFTs, younger age classes are parameterized to have a smaller soil carbon stock. This serves mainly as a preliminary attempt to have cohorts of secondary lands for herbaceous vegetation. Because the directional change of soil carbon largely depends on the vegetation types before and after LUC and on climate conditions (Don et al., 2011; Poeplau et al., 2011), ideally agricultural cohorts from different origins (and age since conversion) should be differentiated, with a origin-specific soil carbon boundary parameterization. However, to avoid inflating the total number of cohorts and the associated computation demand, as a first attempt here, we simply divided each herbaceous PFT into two broad sub-grid cohorts according to their soil carbon stocks and without considering their individual origins. We expect that such a parameterization can accommodate some typical LUC processes, such as the conversion of forest to cropland where soil carbon usually decreases with time, but not all LUC types (for instance, soil carbon stock increases when a forest is converted to a pasture).*"

We further added the following sentences in Sect. 2.3.2: "*Overall, this feature of separating herbaceous MTCs into multiple cohorts is coded more as a "place holder" for the current stage of model development rather than having solid scientific significance. Fully tracking soil carbon stocks of different vegetation types and their transient changes following land use change would require a much larger number of cohorts than that used in this study.*"

Finally, as the differences in land turnover emissions between the two simulations with and without sub-grid cohorts are mainly driven by sub-grid secondary forest dynamics, the influence of errors in setting herbaceous cohorts is expected to be small.

Technical:

It should be made clear earlier in the paper that the terms "shifting cultivation" and "turnover" are used interchangeably.

This is a good point. We put at the end of the revised introduction the following sentence: "*Hereafter, we will use the terms 'shifting cultivation' or 'land turnover' interchangeably as they refer to the same process in the model — bi-directional equal-area land transitions between two land use types*".

Please repeat the main quantitative findings of the study in the conclusions. In some cases letters are swapped in the subscripts.

We repeated the main quantitative findings in the conclusions of the revised manuscript and subscripts are double-checked.

Figs. 4-6 and S7: Please swap the order of the sub-panels from column-wise to row- wise. This is used in Fig. 2 and is much more intuitive.

Following the reviewer's suggestion, we have revised Fig. 5–6 and Fig. S7 using a row-wise format. For Fig. 4, we have kept the current layout. This figure shows in each row, the simulated $E_{LUC}$ by $S_{ageless}$, the age effect and the concerned LUC area involved for each LUC type. Although we don't have any serious scientific papers to support this, we think to compare maps in a horizontal layout is more intuitive to catch the differences.

Figs. 5, 6 and S7: The order of the geographical regions seems totally random. Please introduce some "around-the-globe"-ordering as in e.g. v.d.Werf et al. (2010). I am not saying, that the authors should adopt the regions from v.d.Werf - just the systematic ordering principle.

Thanks for this good suggestion. The order of regions in these figures is re-arranged in a broad sequence of from the south to the north, and from the west to the east. The presentation of different regions in the main text follows their importance of contribution to the global $E_{LUC}$, and in a sequence of from "the highest emissions" to "moderate emissions" to land sinks in the latter half of the 20$^{th}$ century like in the region of Former Soviet Union.

Figs. 3d-3f, 6, S3, S4 and S6: The unit Mkm^2 is not a valid SI unit (double prefix). Please use "Mill. km^2", "10^6 km^2", "10^12 m^2" or rescale to e.g. "MHa" (which would fit the numbers in Figs. 3 and 6 quite well).

Thanks for this good suggestion. The unit of Mkm^2 has been changed to $10^6 km^2$ in all the figures mentioned by the reviewer.

Fig. 5 vs. 6: It is confusing that Fig. 5 starts in 1900 which Fig. 6 starts in 1800. The only thing mentioned in the paper before 1900 is - as far as I see - the peaks in North America. Does that need to be displayed?

Indeed, the only reason to start the horizontal axes of Fig. 6 from the year 1800 is to show the strong legacy impact on emissions in North America. Following the reviewer's suggestion, we have changed Fig. 6 to have the same horizontal axis range as Fig. 5. The pre-1900 LUC area in North America is still described but without a figure being shown.

Table 2: The main point of this table is the threshold fractions of Bmax used - the ages used for the determination are only relevant for the development stage and thus these are the numbers which should show up in brackets. Please either leave out "x Bmax" (described in the table caption) or add it everywhere - the mixture leaves the table rather confusing. The PFT-numbers are only of model internal relevance and should be removed.

We have adjusted the table to put the age information within the brackets and to put the information of fraction of $B_{max}$ in the main table cells, with the meaning of "x $B_{max}$" being explained in the table caption.

In GMD118 l.477 and l.688 the LUH1 data set seems attributed to Hurtt et al. (2006) while the actual description of the data are in Hurtt et al. (2011).

In these two places the description of residence time of shifting cultivation (15 years) is cited from Hurrt et al. (2006). We now use exclusively Hurrt et al. (2011) in the revised gmd-2018-118 paper.

The initial nomenclature is in my opinion more confusing (through unnecessary abstraction of rather simple expressions) than helpful and could be removed.

We would like to keep this nomenclature if it is allowed according to the journal policy, with the hope that it can facilitate the reading process for the readers without a specific land use change research background.

My personal opinion is that supplemental material should be kept at a minimum. For this paper this implies that the description of the backcast of the LUH1 data should rather be an appendix to the paper - or to GMD118 if the method was also applied here. Raw figure data should rather be "available upon request" than put in the supplement.

We believe the suggestion to put in an appendix the back-casting of historical land cover maps is better than putting them in the Supplement if the journal policy allows. We will check with the editorial staff of the journal on this. Several papers have put the raw data in the Supplement and we followed them, but this might not be compulsory. We will also check with the editor on this.

[revised manuscript text omitted]

---

## Author Response (AR2)

**Responses to the reviewer's comments**

In this revision, Yue and co-authors improved the manuscript a lot, among others by sensibly incorporating answers to the points of criticism from the reviewers. In particular I have be focusing on their answers to my own reviewer comments. In general the (my) reviewer comments have been very nicely addressed in the revised manuscript. In few cases answers have been given only in the final author comments in the interactive discussion. Since, however, these answers were of rather technical nature, which would rather tend to obscure than to clarify the manuscript, I find this approach justified. With one exception (below) my comments to the revised manuscript are either praise or of technical nature.

General comments:
The separation of this paper from the accompanying paper is – at least from the point of view of this paper – well implemented.

The inclusion of the ensemble of runs with different priority to the different cohorts very nicely highlight the influence on the assumptions done by the authors and let the reader judge the solidness of the results.

The results presented in the previous version of the manuscript suggested to me, that there was significant non-linear interactions between land turnover and wood harvest. By including the S2b experiment series the authors have demonstrated that such non-linear interactions only play a marginal role and thus the order of process inclusion is rather unimportant.

We greatly appreciate the reviewer's efforts to review our revised manuscript. Besides addressing the reviewer's specific comments below, we carefully went through the whole text and tried to improve the overall manuscript quality.

Specific comments:
L.75-77: Sentence is hard to read. Perhaps leave out "depending on different models and assumptions" which I assume is clear from the context.
Done.

L.94: I am no native speaker, but I am rather sure that "explicit representing" is not grammatically correct. Either "explicitly representing" or "explicit representation of" would work.
We changed to "explicit representation"

L.95, 181 (and elsewhere): I do not really like the expression "time length" but acknowledge, it is not easy to find a good alternative. Maybe anyhow give it a try?
We tried to change it simply to "years".

L. 154: Wrong reference style: "... in Piao et al. (2009a)" instead of "... in (Piao et al., 2009a)".
Done.

L. 155: Guess "are" should be "were".
Done.

L. 206-207 vs. l. 211-212: In the first sentence it is stated that shifting cultivation and wood harvest are (primarily) targeting different cohorts, in the second it is stated that these two processes are treated identically. Please clarify.
This is clarified. In lines 206–207, "wood harvest" should have been "secondary forest harvest", which is now corrected. We removed the sentence of "*Secondary forest wood harvest follows the same rule as shifting cultivation regarding on which forest cohorts to clear.*" The rule for primary forest harvest remained unchanged as described in lines 212–215.

L. 214: No comma after "ones".
Done.

L. ~318: For completeness, also FFire should be explained.
We inserted "$F_{Fire}$ *for carbon emissions from natural and anthropogenic open vegetation fires*".

L. 362-366: Something is wrong with the first part of the sentence.
We changed the first half to: "*We acknowledge that using such static woody biomass boundaries cannot ensure a forest of an exact given age to be cleared in the transient simulations,*". Hope this is clearer.

L. 370: The "- globally" disturbs the reading. Perhaps "in general" (without dash) would fit better? This would also suggest that the uncertainty not only has a geographical but also a temporal component (change of cultivation practices over time at a fixed location).
We followed the reviewer's suggestion as the way suggested conveys more clearly what we intend to mean. The other "in general" in the former half this sentence has been removed.

L. 381: "MTC" is not defined in this paper. I guess for the purposes here, "PFT" would be the right substitution though a more thorough discussion is in the accompanying paper.
Thanks for pointing this out. "PFT" would fit here and we changed to "PFT".

L. 433: "magnitude of areas" is strange, just use "areas".
Following the reviewer's suggestion, we removed "magnitude of".

L. 441: Due to the insertion of the sentence about the S2b experiments, it is no longer obvious which "two simulations" are meant.
We changed "the two simulations" to "$S_{age}$ and $S_{ageless}$ simulations" for clarity.

L. 447: ", dominated by tropical regions" can be left out since this is a part of the gross transition definition introduced by the construction of the LUH1 data set.
We agree on that there is no need to repeat "dominated by tropical regions" as spatial limit of shifting cultivation has explained in the methods section. The suggested revision is done.

L. 533-534: Figure references are confusing and partly wrong. As far as I can see, in the first bracket, "5d" should be "4d", a "(Fig. 5c)" is missing after "harvest" and "... loss (Fig. 5c, 6d)" should be "... loss (Fig. 5d)".
Sorry for this confusion. This is now cleaned and the whole paragraph double-checked.

L. 547-548: Consider reformulation of "the several beginning decades of the 20[th] century".
We changed to ", which yields a high emission legacy for the beginning years of the 20[th] century".

L. 584: "Table 4" not "Table 3".
This has been changed.

L. 615: Guess "as" should be left out.
Change is done.

L. 653-659: Somehow the numbers stated suggest that the authors claim that 510 < 505 which – I guess – is not what they mean. This may happen either because it is not stated for when the data set of Avitabile et al. (2016) is valid (insufficient definition of "contemporary") or because they actually compare to the values from their S3 experiment which they don't state instead of the S0 results stated. Should be clarified.
We compared the Avitabile et al. (2016) with the contemporary S3 simulation, which has the roughly the same biomass as the year of 1500 by the S0 simulation. We modified relevant texts to: "The simulated contemporary global biomass in the S3 simulations, where all three LUC processes are included, remains almost the same as the 1500 value by the S0 simulation. So the $E_{LUC}$ basically balances out what would have been gained in the global vegetation biomass due to environmental changes."

L. 661-679+Fig. S10: Major point:
1) Still I find that this is a lot of pseudo-quantitative work on a very speculative basis which is better left out.
After reflection, we agree with the reviewer that this section is a little far from the main message of the paper and a lot uncertainties exist in such a correction. Therefore we agree to remove this. In the revised text, we removed the last two paragraphs of Sect. 4.2, and merged the first two paragraphs in Sect. 4.1. Fig. S10 and Table S2 are removed from the Supplement Material. Nonetheless, we answer the reviewer's questions as below.

2) Looking closer at Fig. S10 I can't make the numbers match: In South & Southeast Asia the figure suggest an underestimate of ~70 Pg(C) but in the global this seems only to be ~50 Pg(C) despite also several other regions show larger underestimates, while only tropical Africa overestimates (by only a few Pg(C)). The sum of the reconstructed biomass stock for the different regions also doesn't seem to match the global number. Summing from the regional panels of the figure, I get the reported global biomass to ~465 Pg(C), compared to <400 Pg(C) in the global panel. This – on the other hand – could explain the apparent discrepancy of the underestimations which would fit quite well using 465 Pg(C) as the global value. It is unclear to me if resolving these discrepancies would change the numbers in lines 677-678.
Many thanks to the reviewer for having checked carefully the numbers in Fig. S10. Our responses are as below:

(1) Regarding the seeming inconsistencies between sum of regional values and the global one in 1901 biomass in Fig. S10, it turns out that the observation-based global biomass in 1901 somehow has a wrong number in the original figure (390 Pg C). The correct one should the sum of all regional biomasses (455 Pg C). This is the same as in the panel "Globe" in Figure 3, on Page 5050 of Li et al. (2017). Now with this correct global observation-constrained biomass in 1901, the correct version of Fig. 10 should be as below:

[Figure]

(2) Correcting the issue in global 1901 biomass does not alter much in values presented in lines 677–678. There it is said "*The biomass-corrected global cumulative $E_{LUC}$ for 1850–2005 are 174–207 Pg C for the $S_{ageless}$ simulation, and 161–194 Pg C for the $S_{age}$ simulation (Table S1),*". Actually, a range is given here because the sum of different regional values is bigger than what is obtained by applying the correction directly over the globe. This difference exists because of two reasons. First, the emergent linear constraining relationship reported in Li et al. 2017 is empirical over each region and the globe, therefore it's reasonable that the regional sum and the global value do not match perfectly. Second, a wrong global 1901 biomass value is used, and this has made the mismatch much bigger. When we used the correct global observation-constrained biomass of 455 Pg C, the mismatch between the regional sum and the global value of $E_{LUC}$ after correction is only ~5 Pg C. Updating the values in lines 677–678 with those obtained by only applying the correction at the global scale, we get the biomass-corrected $E_{LUC}$ for 1850–2005 as 198 Pg C for $S_{ageless}$ and 185 Pg C for $S_{age}$ simulations, respectively.

3) How does the biomass reconstruction used for Fig. S10 relate to the values of Avitabile et al. (2016)? As far as I can see, they only can be brought to match if there has been a tremendous increase in carbon stocks over the 20th century (505 Pg(C) above ground (~2010) vs. 465 Pg(C) total (1901)). Please comment on this if you decide to keep the paragraph.

There is no direct relation between Fig. S10 and the values of Avitabile et al. (2016). The reconstructed 1901 global biomass stock in the above figure is based on a linear relationship between current-day and 1901 biomass using DGVM simulations (Fig. 4 in Li et al., 2017) and the current-day biomass map of Carvalhais et al. (2014). The value (455 Pg C) is lower than the current-day value of Avitabile et al. (2016), because only grid cells that underwent historical deforestation were included in the reconstructed 1901 global biomass. However, when we applied biomass-based $E_{LUC}$ correction, we used the simulated total global biomass from ORCHIDEE-MICT instead.

Indeed, I see an increase of about 150 Pg(C) during this period in "my own" model (the CMIP5 version of MPI-ESM), but even this seems insufficient to bring the numbers presented here and in Avitabile et al. (2016) together. The increase seen in MPI-ESM is driven solely from an increase in soil carbon (~250 Pg(C)) and is counteracted by a reduction in the living carbon stocks (~90 Pg(C), due mainly to net decreased forest area). Litter stocks are roughly unchanged.
In ORCHIDEE, $E_{LUC}$ mainly cancels out what's expected to be gained in the S0 simulation from 1501 to 2005. For the $S_{ageless}$ simulations, Both biomass (142 Pg C) and soil C (82 Pg C) increased in the S0 simulation from 1501 to 2005, leading an increase in the total carbon stock of 224 Pg C. $E_{LUC}$ cancels out all increase in biomass in the S3 simulation, i.e., biomass decreased by 14 Pg C from 1501 to 2005 in the S3 simulation, while soil carbon still increased by 36 Pg C during 1501–2005, leaving a slight increase in the total carbon stock of 28 Pg C (with 6 Pg C residing in the wood product pool). The gap between the carbon increase over 1501–2005 between S0 (224 Pg C) and S3 (28 Pg C) yields the cumulative $E_{LUC\ ageless}$ (196 Pg C) reported in the paper.

L. 697: "differed" -> "different"
Done.

L. 702-703: The insert "as in the case where they're represented with a single patch within the model" should be marked as insert either by commas, dashes or brackets.
We inserted two dashes to separate this sentence structure.

L. 702: "they're" -> "they are"
Done.

L. 985-987: Using Bmax for both "maximum biomass" and "equilibrium biomass"?!
They indeed mean the same things. To make it clear, we removed both "Bmax" within the parentheses. In section 2.3.2, we modified one sentence to make this clear: "Next, these ratios are multiplied with the equilibrium woody biomass at each grid cell, *__approximated by the woody biomass at the end of model spin-up,__* to derive a spatial map of thresholds in woody biomass.". The caption of this table has also been improved.

Tables 2 and 3: Probably just due to the draft format, but the alignment of columns and numbers is not very reader-friendly.
We make texts in all the columns except the first one in both tables being center-placed, hoping this can be better.

Fig. 2: At least within the same figure, it would be nice if sub-panel names and titles are consistently placed either inside or outside the panel.
A good suggestion. We modified the upper panel to make all sub-panel names being outside.

Fig. 3 vs. Fig. 2, 4, 5, S3 and S8: It is also not very reader-friendly that the ordering of the sub-panels of the figures is different between the figures (column-wise vs. row-wise).
Fig. 3 is corrected to be row-wise and the citations in the main texts have been updated.

Fig. 2S, y-axis: Here an "Mkm2" survived.
This has been corrected.

**Other modifications:**

We removed the line of "Houghton (in prep)" from Fig. 1 as the data are retrieved from Hansis et al. (2015), but the citation is not provided there. The most recent publication by Houghton and Nassikas (2017) on global $E_{LUC}$ does not include gross land use change. Because we cannot provide a proper citation for this data source, we decide to remove it from the Fig. 1. This does not affect the interpretation of that figure and relevant conclusions.

References:
Carvalhais, N., Forkel, M., Khomik, M., Bellarby, J., Jung, M., Migliavacca, M., Mu, M., Saatchi, S., Santoro, M., Thurner, M., Weber, U., Ahrens, B., Beer, C., Cescatti, A., Randerson, J. T. and Reichstein, M.: Global covariation of carbon turnover times with climate in terrestrial ecosystems, Nature, 514(7521), 213–217, doi:10.1038/nature13731, 2014.

[revised manuscript text omitted]